# Long-term cargo tracking reveals intricate trafficking through active cytoskeletal networks in the crowded cellular environment

Jin-Sung Park[1], Il-Buem Lee[1], Hyeon-Min Moon[1], Seok-Cheol Hong [1,2] & Minhaeng Cho [1,3]

A eukaryotic cell is a microscopic world within which efficient material transport is essential. Yet, how a cell manages to deliver cellular cargos efficiently in a crowded environment remains poorly understood. Here, we used interferometric scattering microscopy to track unlabeled cargos in directional motion in a massively parallel fashion. Our label-free, cargo-tracing method revealed not only the dynamics of cargo transportation but also the fine architecture of the actively used cytoskeletal highways and the long-term evolution of the associated traffic at sub-diffraction resolution. Cargos frequently run into a blocked road or experience a traffic jam. Still, they have effective strategies to circumvent those problems: opting for an alternative mode of transport and moving together in tandem or migrating collectively. All taken together, a cell is an incredibly complex and busy space where the principle and practice of transportation intriguingly parallel those of our macroscopic world.

Intracellular cargo transport plays a critical role in maintaining the essential functions of living cells[1,2]. For cargo transport, vesicles are constantly formed as a container at the cell membrane, endoplasmic reticulum (ER), and Golgi apparatus[3,4] and transported by motor protein-driven activity on cytoskeletal highways[5,6]. During this transport, cargos exhibit rich and intricate dynamic events such as directional movement, intermittent pausing, turn-around, and road change at the intersection of cytoskeletal highways[7,8], similar to manufactured vehicles operating in the urban road network (Fig. 1a). It is, however, still unclear how these cargos overcome general traffic problems such as road blockage and traffic congestion in the heavily crowded cellular environment[9,10].

Our current understanding of cellular transport has been brought about by fluorescence-based imaging techniques. The specificity, sensitivity, multiplicity, and super-resolution capability of fluorescence microscopy are critical assets as a powerful tool for cell study. Various issues about cargo transport have been addressed, including motor protein dynamics in cytoplasmic environments[11,12], vesicle transport in 3D[8], cargo dynamics at cytoskeletal junctions[7,13],

interactions of cytoskeletal network, and organelles in transport at high spatio-temporal resolutions[14,15]. Fluorescence-based methods are, however, fundamentally limited in long-term live-cell imaging due to photobleaching. Besides, they leave unlabeled cellular constituents invisible, missing all the information about the local environment except labeled target molecules, even though the unlabeled majority likely dictates the apparent behavior of target molecules.

Here, we report the universal feature of intracellular traffic flow, revealed by simultaneously tracking many unlabeled cargos transported along cytoskeletal highways in a highly parallel fashion and for an indefinitely long time using the interferometric scattering (iSCAT) microscopy. The label-free and high-speed imaging by iSCAT allows us to capture the quantitative and physical details of cargo transports and to acquire a realistic picture of nanoscale logistics within living cells. The enormous amount of cargo localization data ($>10^8$) acquired at 50 Hz throughout the entire observation time (~30 min) enables us to reconstruct the fine architecture of cytoskeletal meshwork and visualize the temporal evolution in traffic flow along the active cytoskeletal highways. Intriguingly, cells intrinsically have an efficient transport strategy to

[1]Center for Molecular Spectroscopy and Dynamics, Institute for Basic Science, Seoul, Korea. [2]Department of Physics, Korea University, Seoul, Korea. [3]Department of Chemistry, Korea University, Seoul, Korea. e-mail: hongsc@korea.ac.kr; mcho@korea.ac.kr

avoid an intracellular traffic jam by forming a train of cargos or collectively moving in the same direction, closely resembling our daily life.

## Results

### Cargos in directional motion are tracked in a complex cellular environment via iSCAT microscopy

As a notable label-free imaging technique, the iSCAT microscopy has drawn much attention due to high detection sensitivity[16–19], ultrafast tracking of dynamic nanoparticles[20,21], 3D imaging capability[21–24], and biological applications towards live-cell imaging[25–28]. To identify cellular vesicles and learn how they appear in our home-built iSCAT microscopy (Supplementary Fig. 1), we fed COS-7 cells with 20-nm fluorescent polystyrene (f-PS) beads and visualized them with fluorescence and iSCAT microscopies simultaneously (Fig. 1b–g).

In the static background-removed iSCAT (SBR-iSCAT) image (Supplementary Fig. 2a, c), various subcellular structures are visible with enhanced contrast (Fig. 1c). Besides filamentous structures, large and small globular objects are easily identifiable (yellow and red circles, respectively; Fig. 1c). Filopodia are also clearly observed at the cell periphery (white arrows; Fig. 1c). However, identifying nanoscale cargos amid the complex cytoplasmic environment poses a significant challenge. To selectively detect the scattering signals of the cargo, we leveraged the dynamic properties inherent to the cargo's motion. A notable characteristic of cellular cargo transport is its (bi-)directionality, punctuated by intermittent pauses. Other cytoplasmic constituents appear relatively static or randomly jiggling. To discern mobile cargos from such static objects and sort out directional cargos from random walkers, we employed the time-differential iSCAT (TD-iSCAT) method (Fig. 1d, Supplementary Fig. 2b, d). In the case of a directionally moving cargo, the dipolar appearance of the cargo reveals its moving direction, indicated by the arrow, which points from dark to bright spots (Supplementary Fig. 2d). In contrast, the corresponding SBR-iSCAT images fail to reveal the cargo, as the irregular cytoplasmic background interferes with its detection (Supplementary Fig. 2c). The TD-iSCAT image sequence displays the cargo vanishing instantaneously at $t = 1.96$ s (Supplementary Fig. 2d) when it altered direction, implying a brief pause on a cytoskeletal track. The observed separation between the bright and dark spots represents the displacement of the cargo over the time interval $\Delta t$. The cargo's movement resembles that of a manufactured vehicle, as it slows down, stops, turns around, and speeds up in the opposite direction (Supplementary Fig. 2d). For backward motion, cellular cargos have two options: engaging motor proteins with opposite directionality on the same track or, less likely, using another track with opposite polarity.

Among f-PS beads identified by fluorescence microscopy (Fig. 1e), only eight of them are co-localized in the TD-iSCAT image (Fig. 1d). The trajectories of 32 f-PS beads were acquired by tracking their fluorescence signals at 10 Hz (Fig. 1f). These trajectories could be classified into two distinct types, directional or Brownian. From the representative data in each type (Fig. 1g, Supplementary Movie 1, 2), the diffusion exponents (α) that are the slopes of the MSD (mean squared displacement)-vs.-time graphs are approximately 1.5 and 1 for directional and Brownian motions, respectively, reflecting their markedly different dynamic characteristics (Fig. 1h). Interestingly, we found that TD-iSCAT imaging only captures f-PS beads in directional motion: the insets (Fig. 1g) show that a Brownian f-PS bead, sharply imaged in the fluorescence channel, is obscure in the iSCAT image while a directional f-PS bead with dimmer fluorescence is detected, as shown in the lower and upper insets, respectively. Thus, intracellular cargos are the main target of TD-iSCAT imaging because many native non-fluorescent cargos without containing f-PS beads are readily visualized by TD-iSCAT as indicated by orange circles (Fig. 1d). Although TD-iSCAT imaging may lose track of cargos during pauses, the concurrent use of SBR- and TD-iSCAT imaging allows for continuous, or near-continuous, tracking of the cargos, as demonstrated in Supplementary Fig. 3 and Supplementary Movie 3.

### Cargo-localization iSCAT microscopy reveals the underlying active cytoskeletal highways

As described above, iSCAT directly captures the dynamic feature of individual cargos transported along cytoskeletal filaments without external labels. As a high-speed and time-unlimited label-free imaging technique, iSCAT produces a large amount of imaging data and thus provides a unique opportunity for statistical analysis. We note that cargos in directional motion are confined on cytoskeletal tracks, and the cargo location is a faithful proxy for a cytoskeletal foothold with the cargo size representing the error in the transverse dimension. To calculate the localization precision for a single cargo identified by TD-iSCAT images, its physical dimension was measured from the complementary SBR-iSCAT image. The bright or dark spots in the SBR-iSCAT image, corresponding to the dynamic cargos observed in the TD-iSCAT image, were analyzed using 2D-Gaussian fitting as shown in Supplementary Fig. 4. The average size of a cargo measured was $393 \pm 62$ nm (Supplementary Fig. 5). For bright and dark spots, exhibiting signal-to-noise ratios (SNR) of 6 and 4, respectively, the centers of their Gaussian peaks can be localized with a precision of within 10 nm and 15 nm, respectively.

By collecting a vast number of cargo locations, one can reconstruct the underlying cytoskeletal network. We define cargo location by the center of the bright spot using the MOSAIC ImageJ plugin[29]. We visualized the cytoskeletal network in lamellar and lamellipodium in a COS-7 cell, which was reconstructed from ~ 10-million localization points obtained from ~ 90,000 consecutive frames taken at 50 Hz (Fig. 2a). The procedure of cytoskeleton reconstruction is also shown in Supplementary Fig. 6a, b. The fine architecture of cytoskeletal networks was previously reported by fluorescence super-resolution microscopic imaging[7]. Such super-resolution images were obtained by point accumulation of fluorescent spots originating from photoswitchable fluorophores directly labeled to cytoskeletons, detecting the whole fluorophore-labeled cytoskeletal material. On the contrary, the cargo-localization iSCAT collects the positional information of individual mobile cargos, capturing the cytoskeletal ropeway selectively along which cargos traveled during the observation time (Fig. 2a, b). As expected, cytoskeleton highways are extended from the cell body (upper-right corner) to the cell boundary. These parallel highways are merged and spanned transversely along the cell boundary (Fig. 2a). Their long, straight shape implies that they are built of stiff microtubules. From the curvature-based approach reported recently[30], we were able to determine the value of persistence length (ξ) of cellular microtubule from its reconstructed shape, which was several hundred micrometers and consistent with previously measured values under in vivo conditions (Supplementary Fig. 7)[30,31]. The direction of cargo movement is designated by the polarity of microtubule: the (+) and (−) ends of the microtubule refer to the direction toward the cell boundary and cell body, respectively. The traffic (or cargo-localization) density map (Fig. 2c) shows the total counts of cargo localization at each pixel through the whole recording of about 90,000 frames ($\Delta T \sim 30$ min). Since ER, the continuous membranous organelle surrounding the nucleus, is the departure station of vesicles carrying just synthesized proteins, the traffic density in the cell body is higher than in peripheral regions. The trajectory of an f-PS bead drawn by its fluorescence is overlaid by a white line drawn on the traffic density map and well-matched with the main traffic route identified in iSCAT imaging (Supplementary Fig. 6g).

We also analyzed the speed of cargos in directional motion along a cytoskeletal highway (Supplementary Fig. 6). The TD-iSCAT frequently loses track of cargos whenever they intermittently stop (Supplementary Fig. 2d). Thus, it is a delicate task to obtain a long trajectory of cargo without the help of fluorescence imaging of the f-PS bead that co-propagates with the cargo. Here, we selected 9 and 10 cargos moving to the (−) and (+) ends, respectively, continuously tracked over at least 3 s (> 150 consecutive frames) with iSCAT

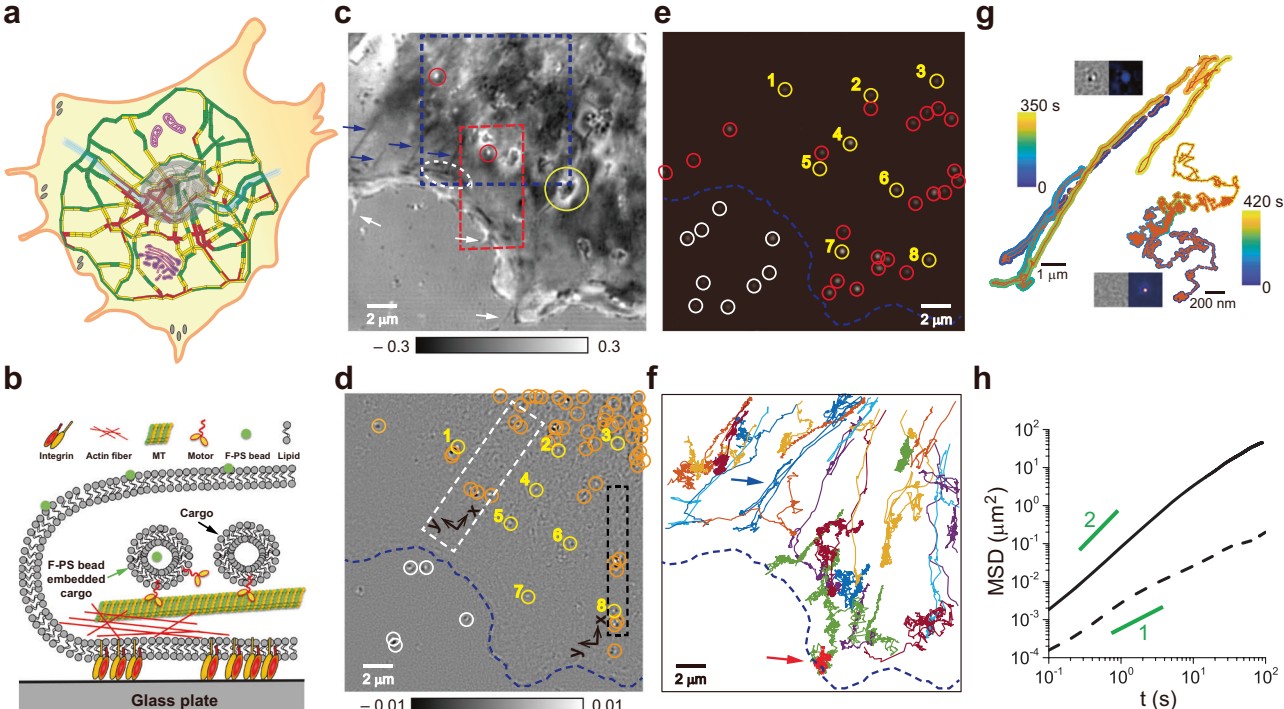

**Fig. 1 | Intracellular cargo transport in a complex cellular environment.**
**a** Schematic illustration of intracellular trafficking based on the traffic map of Seoul, Korea. Colors on each road section indicate traffic congestion (red: heavy, yellow: medium, and green: light). The large gray oval in the center, complex purple structures, and gray spots at the cell boundary represent the nucleus, cytoplasmic organelles, and focal adhesions, respectively. **b** Schematic picture of intracellular cargo transport along a cytoskeletal filament. Cargos bearing an f-PS bead can be detected simultaneously with iSCAT and fluorescence microscopies. **c** SBR-iSCAT image showing intracellular structures in dark and bright contrast (blue arrows and white-dotted oval: filamentous structures and their intersection; yellow and red circles: large and small globular structures; and white arrows: filopodia). **d** TD-iSCAT image of the same area as in **c** to show the locations of moving cargos. In **c**, **d**, the grayscale legends represent iSCAT contrasts. **e** Fluorescence image of the same area showing the locations of f-PS beads. Yellow circles marked with the same

number in **d**, **e** indicate co-localized cargos and f-PS beads, respectively, signifying f-PS bead bearing cargos. Orange and red circles mark spots observed in one channel only (orange in **d**: cargos, and red in **e**: f-PS beads). White circles in **d**, **e** show spots outside the cell. **f**, Trajectories of 32 different f-PS beads tracked at 10 Hz by fluorescence microscopy. **g** Trajectories of two f-PS beads representing (bi-)directional or Brownian motions, indicated by blue and red arrows in **f**, respectively. Insets: TD-iSCAT (left) and fluorescence (right) images of beads in directional (upper) or Brownian (lower) motion. Color legends indicate cargo time points along trajectories. **h** MSD-*vs.*-time curves of the particles in **g**. Solid and dashed lines represent MSD curves for directional and Brownian particles with diffusion exponent of α ~ 1 and 1.5, respectively. Lines with α = 1 and 2 are shown in green. In **d**–**f**, the cell boundary was indicated by a dashed blue line. See Supplementary Fig. 2 and Supplementary Movies 1 and 2. Source data are provided in a Source Data file.

(Supplementary Fig. 6c, d, respectively). Before evaluating the 'instantaneous' speed of cargo, we first acquired the 'moving-averaged' position to suppress noise in speed measurement. The *x* and *y* positions of cargo at 50 consecutive time points in a raw trajectory were averaged to yield the position of the cargo in the 50-point time-averaged (1 s) trajectory (Supplementary Fig. 8 and Supplementary Movie 4). Then, we calculated the instantaneous speed from the displacement made by the cargo for 0.02 s and collected such data to get the distributions of instantaneous speeds of cargos moving to the ( − ) and (+) ends (Supplementary Fig. 6e). From the distributions, the average speeds of cargos moving to the ( − ) and (+) ends were 0.45 ± 0.19 μm/s and 0.53 ± 0.28 μm/s, respectively. Dynein and kinesin motors are the key players responsible for transporting cargos toward the ( − ) and (+) ends of a microtubule, respectively[32]. The speeds of those proteins have been measured using single-molecule techniques in vitro and in vivo, varying widely from 0.02 to 2 μm/s[33,34], indicating that our estimates are within the range of measured values.

Next, we investigated the temporal evolution of the traffic of intracellular cargos, examining a sequence of traffic density maps, each acquired at the time interval of 60 s and integrated for *ΔT* = 180 s (Fig. 2d, Supplementary Movie 5). At *ΔT* = 0 − 180 s, a packet of cargos was visible in the red-oval area, moving to the cell boundary. At the intersection of cytoskeletons, this packet was divided into two and driven in opposite directions along the cell boundary. From TD-iSCAT

images, we could recognize individual cargos in a packet and count them (Supplementary Fig. 9). During propagation, each packet was elongated by combining it with additional cargos approaching from the lamellar side, as indicated by red arrowheads in the 3rd and 4th images. Once it reached the extruding edge of the lamellipodium, this traffic flow gradually disappeared in the 5th and 6th images. We also observed that cargo packets converged toward the point from which a filopodium is growing (Supplementary Fig. 10). More traffic packets were formed in yellow-boxed regions (Supplementary Fig. 10b). We found that the two vital traffic routes along the cell boundary appeared as thick dark bands (white arrows) in the SBR-iSCAT snapshot (Supplementary Fig. 10a), revealing the underlying structural element, actin arcs, for directed cargo movement.

To support that intracellular traffic phenomena are microtubule-based cellular events, we imaged a COS-7 cell expressing GFP-labeled microtubules using F-iSCAT microscopy (Supplementary Fig. 11). The cargo-localization density map (Supplementary Fig. 11e) exhibited a remarkable correlation with the fluorescence image (Supplementary Fig. 11d) obtained simultaneously. For instance, intersections of microtubules (circles in Supplementary Fig. 11d) were co-localized with areas of high cargo density in the density map (Supplementary Fig. 11e) and common passageways for cargo transport (intersections of cargo trajectories) (Supplementary Fig. 11f). Further analysis of the trajectories of individual cargos superimposed on the TD-iSCAT snapshot

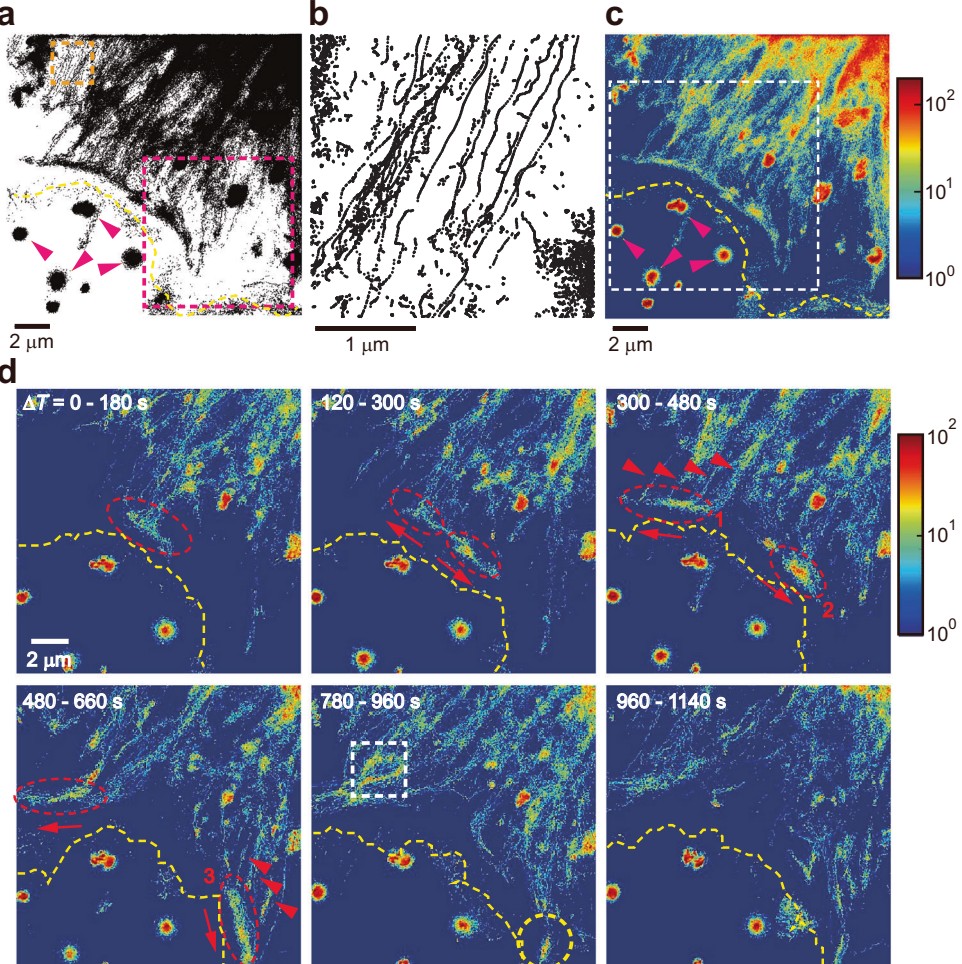

**Fig. 2 | Temporal evolution of cargo flow on active cytoskeletal highways.**
**a** Cytoskeletal network reconstituted by cargo localization performed in TD-iSCAT images (Fig. 1d). About ten-million cargo positions from ~ 90,000 consecutive frames taken at 50 Hz ($\Delta T$ ~ 30 min) were used to make this map. **b** Zoomed-in image of the orange-boxed area in **a**, showing the fine structure of cytoskeletal highways. **c** Cargo-localization density map. **d** Sequence of cargo-localization density maps integrated for 9000 consecutive frames ($\Delta T = 180$ s) taken from the white-boxed region in **c**. Only six characteristic images are presented here. The packet of cargos (red oval in the 1$^{st}$ image) was branched out at the intersection of cytoskeleton highways. Then, these two packets rushed away in the opposite directions to the protruding areas (1$^{st}$ to 4$^{th}$ image). During this process, the traffic density increases as other packets are combined at the intersections of cytoskeleton highways (red arrowheads in the 3$^{rd}$ and 4$^{th}$ images). Numbered red ovals in the 3$^{rd}$ and 4$^{th}$ images indicate the cargo packets, the numbers of which are counted as shown in Supplementary Fig. 9. Once a packet reached the protruding area, it was diminished (yellow circle in the 5$^{th}$ image). In **a**, **c**, purple arrowheads indicate debris fluctuating outside the cell. In **a**, **c**, and **d**, the cell boundary was shown in a dashed line (yellow) for visual guidance. In **c** and **d**, the color legends represent the cumulative cargo localization counts per pixel throughout the entire recordings of 90,000 and 9,000 frames, respectively. See also Supplementary Figs. 6, 9 and Supplementary Movie 5. Source data are provided in a Source Data file.

(Supplementary Fig. 11f) revealed that the microtubule filaments connecting these intersections were frequently used as primary transport routes. When microtubule polymerization was blocked by treatment with 2 µM nocodazole, the traffic flows were significantly reduced compared to the levels observed prior to treatment. (Supplementary Fig. 12 and Supplementary Movie 6 and 7). This observation is consistent with previous studies illustrating that the removal or stabilization of microtubules through drugs can significantly decrease microtubule-mediated vesicle transport[35–37].

Furthermore, we succeeded in uncovering the identity of some cargos by concurrently tracking them in both iSCAT and fluorescence channels. For example, we fluorescently labeled late endosomes (LEs) in a COS-7 cell and identified 14 dynamically transported LEs among others (yellow circles in Supplementary Fig. 13a, c). The trajectories of the LEs were well-overlapped with dense regions of the traffic density map reconstructed from the entire 15,000-frame image sequence, taken over 300 s. Four representative trajectories of the LEs were well-aligned with the cytoskeletal network revealed by cargo-localization

iSCAT (dubbed CL-iSCAT) (Supplementary Fig. 13e, f). See also Supplementary Movie 3.

## Cargos experience heavy traffic at the intersection of cytoskeletal highways

One of the advantages of iSCAT for live-cell imaging is its capability of providing the brightfield image of the entire cellular area, visualizing dynamic events of interest as well as the static subcellular background by applying SBR- and TD-iSCAT image processing methods. In our case, the behavior of mobile cargos can be better elucidated in the context of the local cellular environment, which is invisible in fluorescence imaging. iSCAT imaging reveals that a traffic jam occurs on the cytoskeletal highways in a live cell. We observed that a jammed intersection was temporarily formed (Fig. 3a), and many cargo-like spherical objects were found at this intersection in the zoomed-in SBR-iSCAT snapshot (left in Fig. 3a). Among them, the locations of 5 individual objects exactly matched those of moving cargos identified in the corresponding TD-iSCAT image (right in Fig. 3a). The number of

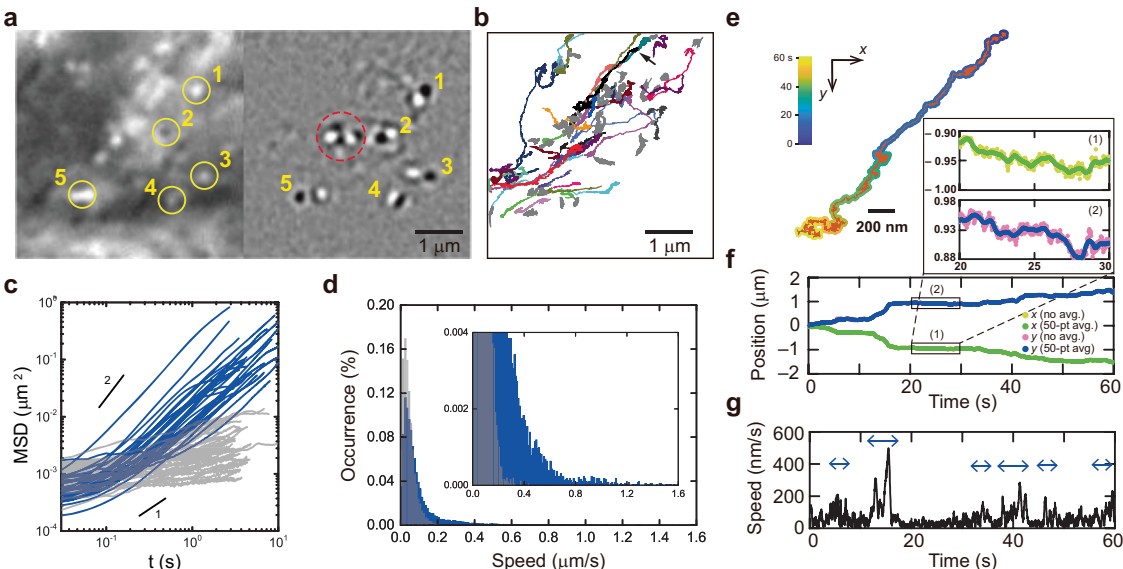

**Fig. 3 | Traffic jam at the intersection of the cytoskeleton. a** SBR-iSCAT (left) and TD-iSCAT (right) snapshots showing several closely located cargos (boxed area drawn at $t = 780–960$ s in Fig. 2d). Both bright and dark spots in the SBR-iSCAT image (numbered yellow circles) are associated with dynamic cargos shown in the TD-iSCAT image. In the red circle in the TD-iSCAT image, a few cargos are closely jammed. **b** A total of 83 trajectories of cargos continuously tracked over 500 frames (10 s) in 9,000 consecutive SBR-iSCAT images (180 s). Among them, the trajectories of 49 cargos with sub-diffusive motion were plotted in gray. The trajectories of 34 cargos showing directional motion were drawn in different colors. **c** MSD-vs.-time curves of cargos traced in **b** (blue: cargos with directional motion; and gray: sub-diffusive cargos). **d** Histogram of 'instantaneous speed' of cargo calculated from the two different groups, directional (blue) and jammed cargos (gray), in **b**. The

'instantaneous' speed was calculated from time-averaged data points (averaged over 50 raw data points for $\Delta T = 1$ s) at the interval of 0.02 s for the raw trajectories drawn in **b**. Directional cargos are dominant over jammed cargos in the regime of high-speed transportation ( $> 0.2$ µm/s) (inset). **e** A representative trajectory of a directional cargo (black trajectory indicated by the arrow in **b**) observed around the jamming area. Color legend is used as in Fig. 1g. **f** $x$ (time-averaged: green; raw: yellow (see inset)) and $y$ (time-averaged: blue; raw: purple (see inset)) positions of the cargo in **e**. **g**, Instantaneous speed of the cargo in **f** calculated from 50-point time-averaged $x$ and $y$ positions of the trajectory shown in **f**. Insets in **f** show the scatter of $x$ and $y$ positions in the raw data with respect to their time-averaged counterparts. See also Supplementary Fig. 8 and Supplementary Movie 4. Source data are provided in a Source Data file.

cargos detected in the SBR-iSCAT images is considerably higher, in this case approximately five times greater, than those detected in the corresponding TD-iSCAT images, which exclusively display moving cargos (Supplementary Fig. 14). A few cargos in the TD-iSCAT image seemed to form a cluster (red circle in Fig. 3a).

Instead of tracking dynamic cargos by using TD-iSCAT imaging, which is unsuitable for locally stalled cargos, we additionally tracked bright spots identified from SBR-iSCAT imaging to investigate the diffusion dynamics of cargos. A total of 83 trajectories, continuously tracked over 500 consecutive frames (10 s) taken at 50 Hz, were overlaid (Fig. 3b). Among them, 34 cargos represented in different colors exhibited directional movement with $\alpha \sim 1.5$ as shown in the MSD-vs-time curve (Fig. 3b, c). Others were locally trapped and exhibited sub-diffusive dynamics with $\alpha < 1$ (gray lines; Fig. 3b, c). The distribution of instantaneous speeds calculated from all trajectories of sub-diffusive cargos shows a positively skewed Gaussian shape with a mean value, $<\bar{v}> = 45.5 \pm 33.9$ nm/s (Fig. 3d). Compared to that of locally trapped cargos, the measured average speed of directionally moving cargos is higher, with $<\bar{v}> = 92.6 \pm 117.7$ nm/s, with a distribution shifted towards higher speed (Fig. 3d). One representative trajectory of a directionally moving cargo (Fig. 3e), however, showed that it was also congested intermittently throughout the observation time (Fig. 3f), and its instantaneous speed, except for short periods of time marked by arrows (Fig. 3g), is indistinguishable from those of sub-diffusive cargos.

### Dynamical phenomena revealed in cargo transport resemble commonplace events in transportation

The superb detection sensitivity of iSCAT allows various dynamic events to be revealed in the process of intracellular cargo transport. Interestingly, we found that some cargos move together as a pair, namely "dimeric cargo" (Fig. 4a). For a long observation time ( $> 130$ s),

cargos in such a dimeric form moved along the same path without being separated. However, its journey was not smooth due to many obstacles scattered along the course. The dimeric cargo moving toward the cell boundary could not move any further after colliding with a large cellular obstacle (time window 'a' in Fig. 4b). After stopping or randomly jiggling for a while, it changed the direction of motion and retraced its path (Supplementary Movie 8). Then, its direction of motion changed once again by a head-on collision with another cargo moving oppositely along the same path (arrow 'b' in Fig. 4b, Supplementary Movie 9). The dimeric cargo stopped again at the surface of the large obstacle (time window 'c' in Fig. 4b), turned around, and then persistently propagated to the cell body (Supplementary Movie 10).

Despite experiencing a bumpy journey, the cargo remained dimeric, suggesting that the dimeric cargos are generally stable. Within a lamellipodial region (Fig. 2c), we monitored at least eight moving dimeric cargos for ~ 30 min (Supplementary Fig. 15), indicating that their existence is not a rare event. Long-term tracking of dimeric cargos was challenging due to their departure from the field of view or signal interference, making it difficult to follow them to their final fate. However, their long-term stability was evident as they all remained bound throughout the entire observation period. In one exceptional case observed in another COS-7 cell, a dimeric cargo split into two monomeric cargos (Supplementary Fig. 16 and Supplementary Movie 11). A cargo trimer was also observed to move together along the same route (Supplementary Fig. 17, Supplementary Movie 12). During translocation, one of the cargos was detached at $t = 210.7$ s, and the rest (dimer) continued to move and turned around at the intersection of the cytoskeleton from $t = 215.7$ s to 216.8 s.

During cellular cargo transport, dynamic events reminiscent of car transportation were observed. One such example was found while monitoring a dimeric cargo. As noted, dimeric cargos are interesting in

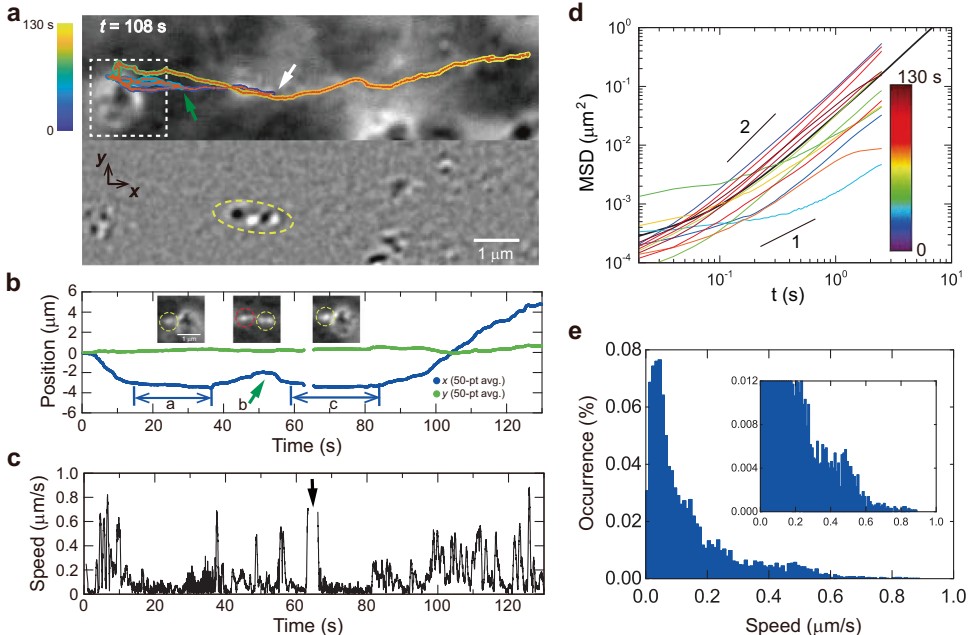

**Fig. 4 | Dynamic events revealed by a directionally transported dimeric cargo.**
**a** Trajectory of a dimeric cargo (yellow oval) showing a directed movement, turn-around, and intermittent pausing (upper: SBR-iSCAT snapshot, and lower: TD-iSCAT snapshot), found in the black-boxed area in Fig. 1d. Here, the white and green arrows indicate the starting point at $t = 0$ and the turn-around point (time point 'b' in **b**) of the dimeric cargo, respectively. Color legend is used as in Fig. 1g. **b** x (blue) and y (green) positions of the dimeric cargo obtained by averaging raw data taken at 50 Hz for 1 s. During the pausing periods ('a' and 'c'), the dimeric cargo (yellow circle in the left and right insets) interacts with a large spherical cytoplasmic structure placed on the right side. At the time point of 'b', the dimeric cargo (yellow circle in the middle inset) collided head-on with another cargo (red circle in the

middle inset) moving in the opposite direction. **c** Instantaneous speed of the dimeric cargo calculated from time-averaged data points (averaged over 50 raw data points for $\Delta T = 1$ s), which was measured with the interval of 0.02 s for the raw trajectory drawn in **a**. When the dimeric cargo was located too close to the large spherical cytoplasmic structure, the automatic tracking program failed to work for a short time indicated by the arrow. **d** MSD-*vs.*-time curves obtained from its whole trajectory (~130 s) drawn in (a) (black) and from each time interval of 10 s (13 curves in different colors as denoted in the color legend). **e** Histogram of the 'instantaneous speed' of the dimeric cargo obtained from (c). (Inset: rescaled histogram emphasizing low occurrence events.) See also Supplementary Fig. 18 and Supplementary Movie 13. Source data are provided in a Source Data file.

their own right and reliable, making them preferred targets for our cargo transport study. Their dumbbell shape allows for clear identification as cellular cargos, distinguishing them from other monomeric cargos in cargo-dense regions. Utilizing fluorescence-combined iSCAT microscopy, we were able to simultaneously detect scattered light from cargos and fluorescence from internalized f-PS beads for specification. This allowed us to identify a dimeric cargo straightforwardly and accurately when associated with an f-PS bead (Supplementary Fig. 18a, Supplementary Movie 13). In this instance, an f-PS bead was identified as part of a dimeric cargo upon close examination using iSCAT imaging. This dimeric cargo exhibited a sporadic journey characterized by frequent stalling, turnarounds, and unsteady movements. Interestingly, the dimeric cargo was put on hold because other cargos crossed the intersection ahead of it (Supplementary Fig. 18f). From an extensive collection of long-term tracking data, we observed that the two cargo modules in a dimeric cargo move in a highly coherent and correlated manner, suggesting that they must be physically bound. We present another compelling piece of evidence for physical binding in a dimeric cargo, which is a 'U-turn' event during its journey (Supplementary Fig. 19). This dimer, composed of two cargo modules, was tracked for ~150 s. The trajectory shows that the dimer makes a U-turn within the green-boxed area (Supplementary Fig. 19a, Supplementary Movie 14). In the sequence of SBR-iSCAT images showing the U-turn, the center locations of the front and rear cargos are indicated by red and blue dots, respectively (Supplementary Fig. 19b). After making a U-turn, the relative orientation of the two cargos was reversed (the front-rear relationship between the two cargos remained unchanged). It indicates that only one of the cargos (front) is actively transported by the associated motor protein on cytoskeleton while the other one (rear) passively trails the active

tractor, hanging from it. Furthermore, the distance between the two cargos (*d*) measured in 2D varied but no more than ~350 nm, which is likely due to the 3D motion of the rear cargo (Supplementary Fig. 19c).

It was recently reported that intracellular cargos move together via 'hitchhiking'. Some cargos literally can hitch a ride with nearby motile cargos already associated with molecular motors[38,39]. As shown above, iSCAT can directly visualize the *outcome* of hitchhiking, multimeric cargo, even when some cargos lack f-PS beads inside. Capturing the moment of hitchhiking is challenging, as it likely occurs at the early stage of cargo transport and near the endoplasmic reticulum (ER), where the cell's interior is too crowded with cargos and other organelles for direct visualization using iSCAT. This concentration of cargos in the ER region increases the likelihood that they will couple together. The hitchhiking process could be a rule rather than an exception to boost the efficiency of logistics by implementing specific adapter proteins bridging different vesicles[40,41]. Thus, it may be programmed for the cell's benefit, which is somewhat analogous to carpooling.

Interestingly, the dynamic events described herein nicely match various occasions encountered during car transportation: a car should detour at roadblocks, wait for other vehicles to cross at the intersection, and come to a stop at the end of the road. It is of note that a complete picture of cellular transportation is now made available by the brightfield iSCAT microscopy.

## Discussion
We demonstrated how iSCAT microscopy could effectively reveal the critical information about intracellular cargo transport. Instead of pursuing chemical contrast using fluorophores, iSCAT utilizes the dynamic feature of transported cargos to discriminate them from

other cytoplasmic objects with similar shapes and iSCAT contrasts. Thus, iSCAT allows us to collect a large amount of data by detecting scattering signals from all kinds of cargos moving directionally along a cytoskeletal track over a long period of time. The massive localization of cargos by iSCAT literally pictures the traffic of cargos and their underlying cytoskeletal highways. Although our cargo-localization iSCAT microscopy constructs the critical visual information with, in principle, the same point localization method generally used in super-resolution fluorescence microscopies[42–45], it reveals quite different features for the following reasons. First, the current super-resolution fluorescence microscopy shows the whole meshwork of cytoskeletal material by accumulating localization points detected from photo-switchable dyes, directly labeled to major components of the cytoskeleton. In contrast, the CL-iSCAT microscopy reconstructs the network from numerous localization points taken from actively transported cargos along the cytoskeletal passage, not from the cytoskeleton itself. Thus, it displays the cytoskeletal pipelines that are being actively used at the moment of observation rather than the whole structure of a cytoskeletal network. Second, as a label-free, high-throughput imaging tool for extensive data acquisition, iSCAT captures the long-time evolution of cargo traffic flows on cytoskeletal highways. In contrast, present-day super-resolution fluorescence imaging is intrinsically limited due to the photobleaching of fluorophores. Recently, several nonlinear vibrational microscopies with chemical selectivity without fluorophore labeling have been introduced for live-cell imaging at a high spatial resolution[46–49]. However, these techniques are insufficient to capture the fast dynamics of cargo transportation in real-time. Thus, the dynamic imaging with the extremely broad bandwidth demonstrated here is unprecedented in fluorescence or vibrational microscopies.

In this study, we determined the locations of intracellular cargos using CL-iSCAT microscopy, combined with image processing methods that extract the dynamic features of moving cargos from the complex cytoplasmic landscape. However, continuous tracking of label-free cargos is limited in duration, as image processing methods tailored for dynamic entities (e.g., TD-iSCAT) may lose track of them when they become momentarily immobile. Additionally, measuring cargo locations can be challenging when cargos overlap with other similar cytoplasmic objects or when their scattering signal is significantly overshadowed by unwanted signals from unspecified cellular environments. While the underlying active cytoskeletal structure can be identified using CL-iSCAT microscopy, the reconstruction process requires a substantial volume of data, comprising thousands of consecutive image frames taken over an extended period (~3 min). As a result, tracking the temporal dynamics of cytoskeleton remodeling on shorter timescales remains unattainable. Furthermore, while state-of-the-art iSCAT technologies enable tracking the Brownian motion of nanoparticles in a three-dimensional space, accurately measuring the variation in their vertical positions within the highly crowded and inhomogeneous cytoplasm remains challenging, due to the presence of numerous optically heterogeneous cytoplasmic objects. Consequently, the cargo-localization density map constructed by CL-iSCAT microscopy should be considered a 2D projection of all cargos moving in 3D within the depth of focus of our microscope. Recently, confocal-type iSCAT microscopies have been developed, offering the ability to investigate nanoscopic structures and dynamics in living cells[50,51]. With its high-resolution detection capability, this imaging technique shows substantial potential in unraveling the intricacies of 3D cargo traffic phenomena along the complex cytoskeleton networks, given its aptitude for detecting nanoscopic displacements in the axial direction.

Our in vivo study showed that intracellular cargos regularly experience heavy traffic throughout the cytoskeletal network. For example, various dynamic events in cargo transport such as pausing, turn-around, and jamming are elucidated by obstacles that block the progress of the cargo, head-on collision with other cargos coming

from the opposite side, or abrupt halt at the intersection of cytoskeletal highways. According to our observations, a cell has inherent strategies for efficient cargo delivery to the destination through a heavily crowded cellular environment. In the short time scale, the traffic of cargos appears to be entangled and stagnant, but in the long time scale, they collectively move in the same direction as a cluster of cargos. In some cases, two or more cargos move together, being directly connected. Our experimental observations strongly support the recent hypothesis of intracellular transport by hitchhiking as one strategy to increase the overall transport rate[38,39]. As summarized above, the iSCAT approach would provide the unique opportunity to investigate the time evolution of nanoscopic cellular constituents in cytoplasmic context at high spatiotemporal resolution.

## Methods
### Cell culture procedures
COS-7 cells, monkey kidney fibroblast cell line, were cultured in a 35-mm confocal dish (SPL, Korea) pre-coated with 0.1 mg/ml poly-D-lysine (P7886, Sigma-Aldrich, USA) for good cell attachment. The seeding density of COS-7 cells was $3 \times 10^4$ cells per dish. Then, cells were maintained at 37 °C in a humidified 5% $CO_2$ and 95% air atmosphere in Dulbecco's modified eagle medium (DMEM) (Gibco, USA) supplemented with 10% fetal bovine serum (FBS, Gibco, USA) and 1% penicillin/streptomycin (Gibco, USA) for 1 day. Before iSCAT imaging, 20-nm f-PS beads were seeded into the sample dish and incubated for 6 h. It is known that f-PS beads incubated with COS-7 cells are spontaneously internalized, trapped in vesicles, and then actively transported along cytoskeletal networks by motor proteins, as schematically depicted in Fig. 1b. Thus, directional cargos containing f-PS beads are likely intracellular vesicles. For live-cell fluorescence imaging, 2 μl of BacMam 2.0 reagent per 10,000 cells were added to COS-7 cells in media. We used two different BacMam 2.0 reagents to label tubulin (C10611, Invitrogen, USA) and LEs (C10588, Invitrogen, USA). Cell samples were incubated for at least 16 h at 37 °C. For (F-) iSCAT imaging, the sample dish was loaded into a mini-incubating chamber (Chamlide, Live Cell Instrument, Korea) mounted on the piezo stage (MZS500, Thorlabs, USA) to maintain the temperature and pH in the sample dish during a long-term iSCAT live-cell imaging.

### iSCAT setup
The fluorescence-combined iSCAT microscopy setup is illustrated schematically (Supplementary Fig. 1). A continuous-wave diode laser (OBIS-FP-647LX, Coherent, USA) illuminates the sample area by raster scanning a weakly focused beam with a two-axis AOD (DTS-XY400-647, AA optoelectronics, France). The telecentric lenses (T1 and T2, $f = 500$ mm, AC254-500-A, Thorlabs, USA) are used to image the input beam onto the back focal plane of the high numerical aperture (NA) oil immersion objective lens (100 × UPlanSApo, 1.4 NA, Olympus, Japan). The reflected and the scattered lights from the sample are collected by the same objective, reflected by the beam splitter (BS, BSW-10R, Thorlabs, USA), and imaged with the tube lens (TL1, $f = 750$ mm, AC254-750-A, Thorlabs, USA) onto an sCMOS camera (PCO.edge 4.2, PCO, Germany). The field of view is about $23 \times 23$ μm². In order to detect fluorescent signals from f-PS beads, two dichroic mirrors (DM1 (FF495-Di03, Semrock) and DM2 (FF552-Di02, Semrock)) were introduced between the objective lens and the beam splitter. DM1 is used to reflect the excitation beam into the main beam path, and DM2 is used to reflect the fluorescent signal out of the beam path and into the detector. The light from an LED (SOLIS-3C, Thorlabs, USA) filtered by an optical bandpass filter (F1, FF02-482/18-25, Semrock, USA) is used to excite f-PS beads, and the fluorescent signal from them, selected by DM2 and an optical bandpass filter (F2, FF02-520/28-25, Semrock, USA), is projected onto an EMCCD via a tube lens (TL2, $f = 400$ mm, AC254–400-A, Thorlabs, USA). The resulting field of view is about $32 \times 32$ μm².

## SBR-iSCAT image processing

In principle, the iSCAT image is obtained by combining the scattering field from intracellular target objects with the reference field reflected at the interface between the glass and the culture medium. The intensity on the detector ($I_{det}$) is given by

$$I_{det}(\propto |E_r + E_s|^2) = I_r + I_s + 2\sqrt{I_r I_s}\cos\varphi \qquad (1)$$

where $E_r$ and $E_s$ are the reference and scattered light fields, respectively, and $\varphi$ is the relative phase between the two fields. As the scattering intensity ($I_s$) is negligible for small objects compared to the reference intensity ($I_r$), the third interferometric cross term becomes the leading contribution from scattering objects. The iSCAT contrast ($C$) is defined as

$$C \equiv \frac{I_{det} - I_{back}}{I_{back}} = \frac{I_{diff}}{I_{back}} = \frac{2|s|\cos\varphi}{r} \qquad (2)$$

where $I_{back}$ and $I_{diff}$ are the intensity by background and its difference from $I_{det}$, respectively, and $r$ and $s$ are the reflection and scattering coefficient, respectively.

The raw images from iSCAT inevitably include static non-uniform background caused by uneven illumination or spurious speckles due to back-reflections by or defects on optical elements placed along the beam path, which deteriorates the quality of raw images. To eliminate such a time-invariant background noise, we employed the static background removed (SBR)-iSCAT image processing method (Supplementary Fig. 2a, c). First, a temporal-median cell image was generated by averaging 10 consecutive raw images ($\Delta t = 0.2$ s) taken at 50 Hz to reduce the level of shot noise fluctuations in iSCAT snapshots. Then, we obtained an SBR-iSCAT image by subtracting a temporal-median background image from the temporal-median cell image to obtain the differential image and dividing the differential image by the background image, as written in Eq. S2 above. The temporal-median background image was averaged for a stack of ~ 3,000 images obtained from a cell-free zone on a glass substrate. The quality of image is significantly improved thereby, and various cellular structures are clearly revealed with enhanced contrast by merely removing the stationary background with reduced shot noise.

## TD-iSCAT image processing

Even with the enhanced contrast in SBR-iSCAT imaging, it is still difficult to identify intracellular cargos in the cytoplasmic space full of unidentified subcellular objects with similar shapes and contrasts. To capture dynamic cargos from the complex cytoplasmic landscape, we used the time-differential (TD)-iSCAT image processing method (Supplementary Fig. 2b, d) instead, highlighting dynamic cellular features by suppressing quasi-static iSCAT signals from relatively stationary cellular substance. First, we obtained a temporal-median cell image from 20 consecutive frames ($\Delta t = 0.4$ s) taken at 50 Hz. By dividing a temporal-median image by another such image taken after the time interval ($\Delta T$) of 0.2 s, a TD-iSCAT image was obtained. In the image, individual cargos which move faster than 200 nm/s appear as a pair of dark and bright spots, indicating the early and late positions in the time interval averaged over for each image ($\Delta t = 0.4$ s). In contrast, other static or randomly moving cellular entities fade or disappear due to signal cancellation. Cargos slower than 200 nm/s would be self-canceled because their signals fall in the same pixel (1 pixel ~ 40 nm) within the time interval of 0.2 s. In the cases cargos backtrack, however, they would be counted multiple times.

## Cargo localization and tracking via the MOSAIC ImageJ plugin

The detection and tracking of multiple cargos from iSCAT images were achieved by the MOSAIC ImageJ plugin[29]. Although this plugin was developed as an analytic tool for fluorescence microscopy images, we found that it is equally well suited for analyzing cargos captured in both SBR- and TD-iSCAT images. There are three parameter values to be set for cargo identification: particle radius ($r$), cutoff score ($s_{cut}$), and intensity percentile ($I_p$). The value of particle radius was set to be the diffraction limit, $\Delta r' = \frac{1.22 \times \lambda}{2 \times NA}$, where the numerical aperture ($NA$) and $\lambda$ used for iSCAT detection are 1.49 and 647 nm, respectively. Thus, $\Delta r'$ is about 264 nm, corresponding to ~ 7 pixels in our iSCAT setup (1 pixel ~ 40 nm). Accordingly, larger cargos with a radius > 7 pixels were excluded, and two (or more) cargos less than 7 pixels apart could be counted as a single cargo in our analysis. The cutoff score for non-particle discrimination was not applied in our cargo identification and set to '0'. $I_p$ is the parameter to determine which bright pixels are accepted as cargos. While the dynamic cellular features are only highlighted in TD-iSCAT cell images, SBR-iSCAT cell images reveal the whole cellular landscape, including static or randomly moving objects with enhanced iSCAT contrast. To construct the cargo-localization density map, we selectively detect dynamic cargos, each of which appears as a pair of bright and dark spots in TD-iSCAT images, by setting the value of $I_p$ to 0.3 (Supplementary Fig. 20). SBR-iSCAT images were synchronously examined to detect locally stalled cargos in a traffic jammed area (Fig. 3b) or cargos that are intermittently stalled during a long journey (Supplementary Fig. 3) by applying a different value of $I_p$ to 5.

For cargo localization to reveal the underlying active cytoskeletal highways, the spatial coordinates of individual cargos are determined in the TD-iSCAT image with the following localization precision,

$$\sigma^2 = \left[\frac{\left(s^2 + \frac{q^2}{12}\right)}{N}\right] + \left[\frac{8\pi s^4 b^2}{q^2 N^2}\right] \qquad (3)$$

where $s$ is the standard deviation of the point-spread function, $N$ the total number of detected photons of the scattered light, $q$ the equivalent dimension of a pixel in the sample space (~ 40 nm), and $b$ the background noise per pixel[18,52,53]. Considering the full well capacity of the camera (~ 30,000 e⁻) given by the manufacturer, the values of $N$ and $b$ are assumed to be about 7000 and 1500, respectively, measured from a cell-free background area illuminated by 1 mW light. Then, a pair of bright and dark spots revealed in a TD-iSCAT image were separately fitted with the following 2D Gaussian function,

$$f(x) = A e^{-\left(\frac{(x-x_0)^2}{2s_x^2} + \frac{(y-y_0)^2}{2s_y^2}\right)} + B \qquad (4)$$

where $A$ is the height of the peak with respect to the base level $B$, $x_0$ and $y_0$ are the positions of the center of the peak, and $s_x$ and $s_y$ are the standard deviations of the position of a bright (or dark) spot (Supplementary Fig. 4). From our measurements, the value of SNR is ~ 6 (or ~ 4) for a bright (or dark) spot, which allows localization of the center of the Gaussian peak with a precision of ~ 10 (or ~15) nm.

## Instantaneous speed and MSD measurements from cargos' trajectories

To obtain the trajectory of an individual f-PS bead-bearing cargo, the position of the f-PS bead was manually (Fig. 1f) or automatically (Fig. 1g) tracked from fluorescence images taken at 10 Hz. To automatically track the motion of an f-PS bead using the MOSAIC ImageJ plugin, we set $r = 7$, $s_{cut} = 0$, and $I_p = 1$. Trajectories from individual f-PS bead-free cargos (Figs. 3b, e, and 4a) were automatically obtained by applying the plugin to SBR-iSCAT images instead of TD-iSCAT images because the TD-iSCAT scheme fails to track cargos' locations faithfully whenever they are intermittently paused. The instantaneous speeds ($v$) from the trajectories acquired in SBR-iSCAT images (Figs. 3g and 4c)

were calculated from

$$v \equiv \frac{dr}{dt} \sim \frac{r(t + \Delta t) - r(t)}{\Delta t} \qquad (5)$$

where $t$ and $r(t)$ are the time taken and the position of a particle at $t$, respectively. Here, $\Delta t$ is given by $1/f$ (Hz), which is 0.02 s. However, the value of $v$ calculated from a raw trajectory varies considerably, likely for the inaccuracy in finding the center position of cargo due to thermal noise. To suppress noise in speed measurement, we calculated the value of $v$ from a 50-point time-averaged trajectory, the cargo position of which was given by the mean position of the cargo from 50 consecutive raw images (Supplementary Fig. 8). Markedly different from the raw and 10-point time-averaged trajectories, the 50-point time-averaged trajectories show that the baseline or minimum of $v$, which would be the value of speed at the moment of intermittent pause, is close to zero.

The MSD of a particle over a time interval $t$ is defined as the average squared displacement of the particle during the interval, taken over $N$ successive time lags:

$$MSD = \frac{1}{N} \sum_{i=1}^{N} |\mathbf{x}(i\Delta t + t) - \mathbf{x}(i\Delta t)|^2 \qquad (6)$$

where $\mathbf{x}(\tau)$ is the position of the particle at time $\tau$. A long-time particle tracking (> 130 s) necessary for MSD analysis is feasible for a single f-PS bead or an intracellular cargo as demonstrated with fluorescence imaging (Fig. 1g, h) or SBR-iSCAT time-lapse imaging (Fig. 4b, d). In contrast, cargo tracking is severely hampered in the bustling area where cargos frequently jam because the images of cargos are often overlapped there. Thus, we selected a relatively short trajectory of each cargo for MSD analysis and continuously tracked for 150 consecutive frames (or 3 s) in the jammed area (Fig. 3b, c).

## Statistics and reproducibility

The data presented in this work are representative of our experimental results. We conducted iSCAT time-lapse imaging and collected more than 10 different sets of data from different COS-7 cells cultured in the presence or absence of 20 nm f-PS beads at temporally independent periods. Most of the cargo trajectories shown in this work were obtained from the same area (field of view: $20 \times 20 \ \mu m^2$) of a single cell in Fig. 1c unless noted otherwise. From this area, ~ 10-million localization points automatically identified with the MOSAIC ImageJ plugin were used to reconstruct the cytoskeleton network. By applying the cargo-localization method to TD-iSCAT images, we showed that these image data exhibited similar intracellular traffic phenomena including the cargo flow by collective migration (Fig. 2, Supplementary Figs. 9 and 10), the traffic jam in a locally concentrated area of cargos (Fig. 3, Supplementary Fig. 14), and the transport in the form of dimeric cargos (Fig. 4, Supplementary Figs. 15–19). The sequences of cargo density map images shown in Fig. 2d and Supplementary Fig. 10b were obtained from two partially overlapped regions in the same cell, corresponding to the white (Fig. 2c) and purple boxed area (Fig. 2a). Multimeric cargos were observed from 3 different cell culture dishes: one from each culture was dedicated to a specific observation: the long journey of a dimeric cargo (Fig. 4a, Supplementary Fig. 15), the separation of a dimeric cargo (Supplementary Fig. 16), and the formation of a trimeric cargo and its separation (Supplementary Fig. 17). Turn-around events of a cargo were routinely observed (Fig. 4a, Supplementary Figs. 18a and 19a). For the dual-channel F-iSCAT imaging of COS-7 cells with GFP-labeled microtubules (Supplementary Fig. 11), we acquired 6 data sets, taken from 6 different cell culture dishes. The effect of nocodazole (Supplementary Fig. 12) was tested with 4 different cells cultured on different cell culture dishes. For the dual-channel F-iSCAT imaging of COS-7 cells with GFP-labeled LEs, we acquired 2 data sets, taken from two different cell culture dishes. The average speed of cargos moving on cytoskeletal filaments is presented in the form of the mean ± standard deviation and mentioned in the text. All figures were created using OriginPro (Origin lab), MATLAB (MathWorks), or Illustrator (Adobe).

## Reporting summary

Further information on research design is available in the Nature Portfolio Reporting Summary linked to this article.

## Data availability

The authors declare that the data supporting the findings of this study are available within the paper and its supplementary information files. All the raw data generated in this study are provided in the Source Data file. The minimum data sets of the raw iSCAT cell images generated in this study have been deposited in figshare (https://doi.org/10.6084/m9.figshare.23826420). The whole sequences of the raw iSCAT cell images and other relevant data that further support the findings of this study are available from the corresponding authors upon request. Source data are provided with this paper.

## Code availability

The MATLAB code, 'iView.m', used in this study to generate SBR- and TD-iSCAT images from raw iSCAT images has been deposited in figshare (https://doi.org/10.6084/m9.figshare.23826504).

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

## Acknowledgements

This work was supported by IBS-R023-D1 and NRF-2022R1A2B5B01002343 (S.H.); Global Research & Development Center Program (2018K1A4A3A01064272, S.H.) through the NRF funded by the Ministry of Science and ICT.

## Author contributions

Conceptualization: J.P., S.H., M.C. Methodology: J.P., I.L., M.M. Investi-gation: J.P., I.L., M.M., S.H., M.C. Writing – original draft: J.P., S.H., M.C. Funding acquisition: S.H., M.C. Supervision: S.H., M.C.

## Competing interests

The authors declare no competing interests.
