## [Peer Review File · Nature Communications]

Reviewers' comments:

Reviewer #1 (Remarks to the Author):

The manuscript by Park et al. presents an intriguing new approach to the age-old problem of tracking individual cargo particles inside living cells. One of the fundamental recurring questions in this field is whether and to what extent observed particle behavior arises from the properties of the particle itself versus the properties of underlying cellular structures (like the cytoskeleton) in the location it passes through. Fluorescently labeling an individual cargo type obscures the underlying structures and tends to involve limited sampling of a given spatial location. A method such as PIV focuses on specific cellular regions but requires embedding a dense field of labelled particles in the cell. The approach proposed here in principle does not require fluorescent labels with the concomitant photobleaching limitations, and also does not necessarily require the introduction of exogenous particles or overexpression of specific markers in the cell.

As a result, the authors are able to track huge numbers of cargos for relatively long time periods, gathering statistics on many trajectories that pass a given cellular region. A major drawback of the method, however, is that only processively moving cargos are robustly identified and trajectories are truncated whenever the particle stops its processive motion. Another drawback is the inability to distinguish different particle populations -- an inherent tradeoff between studying "all the particles that pass this particular region, regardless of their identity" versus studying "all the particles of a certain type, regardless of their location".

While the method seems a useful and important one, the manuscript suffers somewhat from being largely descriptive in nature. The authors measure quantities like typical cargo speed (not particularly meaningful given it is averaged over a broad population of different organelles). They also describe a few select instances of dynamic events, such as packets of cargos pausing or splitting at specific locations. It is not clear, however, what new insight is gained through these observations, and there is little provided in the way of statistical measurements beyond these one-off 'highlight' events. Some major comments are listed below:

Major comments:

- 1) It was not clear to me from the manuscript what the practical limitations are on the cargo tracking. For instance, what is the range of cargo sizes and speeds that can be robustly extracted by this method? How do the authors overcome issues of crowding with many particles crowded together within a diffraction-limited (or at least very small) zone -- would such situations prevent any cargo from being tracked? How often does that occur? Given that this method does not rely on labeling specific cargos, it

seems surprising that individual particles show up at all given the many particles one would expect to be moving through any given area of the cell.

2) Is it possible to tell the number of cargos involved in a "packet" or "cluster"?

3) Can hitchhiking events be observed directly, in the form of pairs of particles where one is already moving processively before picking up an extra passive hitchhiker? Or are the observed dimers only seen in their dimeric form from start to end of the trajectory?

4) Can statistics on microtubule distribution be extracted from this data. For example, the authors mention a persistence length of 4-8mm for MTs -- the actual bending in vivo has been measured to be much greater, due to active fluctuations (see, eg, Brangwynne, JCB, 2008). Can a persistence length be extracted from the cargo trajectories? How about the location or temporal persistence of MT intersections?

5) It was not clear to me how hitchhiking (in the sense of a direct molecular connection between cargos) could be distinguished from two particles walking close to each other (within the diffraction resolution limit) with similar velocities. Perhaps statistics on how long cargo dimers and trimers persist could be collected? Or statistics on the distribution of packet sizes (how many particles are moving alone vs in larger groups)? These are just suggestions, and not specifically necessary for the paper. But it does seem that the paper would be improved by some quantitative results beyond the velocity measurements which have been previously carried out.

6) Along the same lines, the generalized claims about how cargos move collectively on longer time-scales seem a bit overstated, as only one or two example cases of such motion are shown. How often does this occur? Is this a general feature or an outlier?

Minor comments:

Line 110-111: the MT polarities seem to be reversed. (+) ends should be towards the cell boundary.

Line 113-114: vesicles depart from the ER in the cell periphery as well. Also many cargos presumably depart from the Golgi and from the plasma membrane following the endocytic pathway.

Figure 1c: the dark blue oval is nearly invisible in print

Figure 2c: It is not clear what is meant by "total number of cargos detected at each pixel for 90000 consecutive frames". Is this distinct cargos? Are does the same cargo found in one pixel for multiple frames (either from slow movement or because it backtracks) get counted multiple times?

Figure 4: Can the authors comment on the identity of the large cargos observed (which look beyond diffraction-limited in size)? Are these also vesicles containing fPS beads or are these unrelated particles?

Reviewer #2 (Remarks to the Author):

The work by Park et al. titled 'Long-term cargo tracking reveals intricate trafficking through active cytoskeletal networks in the crowded cellular environment' describes the use of iSCAT microscopy to follow intracellular trafficking of cargo vesicles along cytoskeletal filaments. The authors study Cos-7 cells and use internalized 20 nm fluorescent beads as an additional marker for vesicle cargo transport using fluorescence microscopy in parallel to iSCAT.

The motivation of this work is that iSCAT microscopy offers the benefit of label-free microscopy without the risk of photobleaching, and hence the possibility to acquire long timelapse movies at high imaging frequencies. This enables to follow intracellular dynamics of light-scattering organelles without any selectivity that occurs in typical experimental setting relying on specific labelled proteins.

The authors demonstrate successfully and convincingly that this approach is indeed applicable to follow transport of vesicular cargo along cytoskeletal filaments within Cos-7 cells. Recording the edges of cells and performing two different ways of post imaging processing, i.e. static background removal (SBR) and time-differential (TD), the authors reveal a multitude of cargo shapes and dynamic behaviors. To quantify cargo speeds and trajectories, particle tracking with the MOSAIC toolbox via image J is employed.

Though this work presents an interesting approach to visualize cargo transport within live cells, it is purely descriptive and lacks any controls to support the claims made by the authors about the reasons of the observed dynamics. The number of cells imaged and analyzed is low and deep analysis of interesting cargo transport tracks seems very anecdotal lacking any wider analysis about e.g. the relative occurrence of the different observed trafficking behaviors.

Another aspect that could have been investigated to provide some further insights into cellular trafficking and could have highlighted the potential of the presented methodology would be to use

more sophisticated analysis of the particle tracks. The high temporal resolution would allow to distinguish between phases of diffusive and directional transport and how this relates e.g. to the presence of other cargo vesicles in the surrounding.

Very important or rather essential control experiments are missing, such as using fluorescent markers for specific cellular endosomes (e.g. Rab-5, Rab-7 or Lamp-1) or a check whether the cargos detected are preferentially linked to microtubules or actin filaments. Another set of experiments that could underline that the observed phenomena are related to active transport of vesicles would be to use chemical inhibitors (e.g. blebbistatin, actin or microtubule depolymerizing drugs). Without such control experiments, it remains unclear whether all types of cargo are visualized with iSCAT or whether it is a particular subset. The time window used in the TD analysis could bias the detection for only those cargos that move fast enough, and it would be good if the authors could comment on that.

Taken together, the work in its current form is not suitable for publication in Nature Communications. Below are further minor and technical comments.

Minor comments:

Page 2, line 6: 'paralleling': wording should be changed here. I think, the authors mean something like 'similar to' or 'emulating'

Page 2, line 7: 'unclear how these cargoes overcome...': the traffic analogy might be quite appealing at first sight, but I wouldn't take it too literally.

Page 2, line 10: It's not clear how cell metabolism comes in here. Metabolism, i.e. the conversion of nutrients into energy storage and finally use of the energy to perform tasks is usually linked to ATP and mitochondria activity, not with cargo transport.

Page 2, line 25: localization data (), The authors should clarify what type of data this is. Tracks? Or the individual localization during the entire course of the time lapse?

Page 2, line 36: feeding of Cos-7 cells: some more detail might be of help here describing how beads were internalized into cells. Do you know via which pathway they get internalized? Do they end up in membrane bound vesicles?

Page 3, line 2-3: Overly complicated wording. Use active language and simply say what you did.

Page 3, line 4-5: Please remove the motor vehicle analogy. It is unnecessary here.

Page 4, line 17: TD-iSCAT is not a tracking routine. So how should it lose track? Clarify the wording, please.

Page 5, line 1-2: 'funneled into the building': This is nice, colorful language but maybe not the most suitable for the results section of a scientific article. In addition, the observations here seem very anecdotal, and it is difficult to see what the bigger message is here.

Page 5, line 8-9: It is not clear what the authors want to describe here. What is meant by backdrop? Do you want to say that iSCAT has the benefit of providing both, information about the overall organisation/ build-up of the cell (e.g. by taking the average of the timelapse) and about different dynamic processes (by analysing the differential imaging information)?

Page 5, line 18, use 'unsuitable' instead of 'unusable'

Page 6, line 39-40: It is true that the iSCAT localisations mainly reflects the effective cytoskeletal transport paths, however, tracks that are remodelled, i.e. which change along the time of taking, will not be represented well.

Fig 2: what do the green arrows mean in 'a'? nothing is mentioned about it.

Fig 3: (e): is f) showing the positions of the trajectory in e). If so, the x and y axis graphs should both go from 0 to negative values. Because the trajectory goes from the top right to bottom left corner. If f) depicts another data set, it would be more insightful if the corresponding track is shown in e).

Page 19, line 35 till page 20, line 2 : This has nothing to do in a methods section.

Reviewer #3 (Remarks to the Author):

The authors present a new method for tracing features of the cytoskeleton in live cells. To do so, they combine interferometric scattering microscopy (iSCAT) with differential frame analysis. As cargo

attached to cytoskeletal filaments move, they can be identified on top of an otherwise busy background. In a sense, this clever label-free strategy shares some aspects of fluorescence super-resolution microscopy such as PAINT, except that instead of using a large number of bleaching dye molecules, one exploits indigenous biological cargo. This is only made possible because of the high sensitivity of iSCAT for detecting small nanoparticles.

Although the authors do not address or solve any particular biological question, they show that their method could be used to investigate cargo diffusion and transport, which is a very important and complex matter in cell biology. I, therefore, deem this work worthy of attention of a wider readership. However, there are a number of critical questions that would have to be addressed before the work reaches the required quality for publication in Nature Communications:

1- The authors should refrain from using formulations such as “surprising” or “amazement” when referring to cargo trafficking or other phenomena unless they can provide hard evidence that the observation at hand is surprising, i.e. contrary to what is expected in the literature. They might mean to say “intriguing” instead?

2- The method proposed would only work if the cytoskeletal feature under examination does not change within the measurement time, which seems to be several minutes at times. The authors should make a statement about the limitations of their technique in this regard.

3- The authors do not reveal any critical cell biology information. Instead, their evidence on transport phenomena remains at the speculative level. The authors are encouraged to demonstrate if the observations are representative of what happens on the cellular level.

4- Considering that the traced cytoskeletal filaments cannot perfectly lie in a horizontal plane, the authors should make a statement about the consequences of axial motion, which changes the iSCAT contrast.

5- Although I find the author’s analogies with urban traffic interesting, I believe such continuous emphasis would only be warranted if it could go beyond an anecdotal level and were based on a quantitative statistical analysis. I recommend they remove those references.

6- Page 4, lines 28-32: Can the authors say more about the fact that their measurements are more precise than previous ones?

7- The method would only be truly valuable if it could be implemented without synthetic cargo such as PS beads. The authors should make a clear statement/demonstration of biological information that can be obtained from indigenous cargo.

Manuscript ID: NCOMMS-22-16865A-Z

Title: Long-term cargo tracking reveals intricate trafficking through active cytoskeletal networks in the crowded cellular environment.

Point-by-Point Responses to Referees' Comments

Referee #1

(Overall remark) The manuscript by Park et al. presents an intriguing new approach to the age-old problem of tracking individual cargo particles inside living cells. One of the fundamental recurring questions in this field is whether and to what extent observed particle behavior arises from the properties of the particle itself versus the properties of underlying cellular structures (like the cytoskeleton) in the location it passes through. Fluorescently labeling an individual cargo type obscures the underlying structures and tends to involve limited sampling of a given spatial location. A method such as PIV focuses on specific cellular regions but requires embedding a dense field of labelled particles in the cell. The approach proposed here in principle does not require fluorescent labels with the concomitant photobleaching limitations, and also does not necessarily require the introduction of exogenous particles or overexpression of specific markers in the cell.

As a result, the authors are able to track huge numbers of cargos for relatively long time periods, gathering statistics on many trajectories that pass a given cellular region. A major drawback of the method, however, is that only processively moving cargos are robustly identified and trajectories are truncated whenever the particle stops its processive motion. Another drawback is the inability to distinguish different particle populations -- an inherent tradeoff between studying "all the particles that pass this particular region, regardless of their identity" versus studying "all the particles of a certain type, regardless of their location".

While the method seems a useful and important one, the manuscript suffers somewhat from being largely descriptive in nature. The authors measure quantities like typical cargo speed (not particularly meaningful given it is averaged over a broad population of different organelles). They also describe a few select instances of dynamic events, such as packets of cargos pausing or splitting at specific locations. It is not clear, however, what new insight is gained through these observations, and there is little provided in the way of statistical measurements beyond these one-off 'highlight' events. Some major comments are listed below:

(Reply) We would like to extend our gratitude to the referee for providing valuable and constructive comments. The referee has thoroughly reviewed the relevant research area and succinctly summarized the imaging technique presented in our manuscript, highlighting the major limitations as (i) the imperfect tracking algorithm for cargos that pause intermittently, (ii) the intrinsic difficulty in cargo classification solely based on light scattering, and (iii) the lack of scientific understanding based on detailed quantitative data analysis. In the original manuscript, we discussed the capability of detecting and tracking cargos using two complementary iSCAT image processing methods, static background removed (SBR)-iSCAT and time-differential (TD)-iSCAT (please refer to the first paragraph of the "Cargo localization and tracking via the MOSAIC ImageJ plugin," and "Instantaneous speed and MSD measurements from cargos' trajectories" sections in the manuscript). As the referee noted, TD-iSCAT only captures cargos that are transported along cytoskeletons, eliminating scattering signals from static objects. While SBR-iSCAT may be susceptible to masking of cargos by static structures, it is still

capable of tracking cargos continuously, even if they pause momentarily. As we explained in the second paragraph of the "Cargos experience heavy traffic at the intersection of cytoskeletal highways" section, cargos that are locally jammed in a densely populated area are identified as individual bright spots in in

Rebuttal_Fig. 2 (see **Supplementary Fig. 3** in the revised manuscript). Continuous cargo tracking with SBR- and TD-iSCAT methods. a, Trajectories of a GFP-labeled late endosomal cargo, revealed by fluorescence detection (left, white) and SBR-iSCAT (middle) or TD-iSCAT (right) imaging-based cargo localization. The trajectories by iSCAT are color-coded based on the elapsed time and superimposed on a snapshot taken at $t = 0.0$ s. Color legend is used as in Fig. 1g. b, Comparison of the two trajectories of the cargo obtained by SBR- and TD-iSCAT methods (blue: SBR-iSCAT; red: TD-iSCAT) highlighting the complementarity of the two methods for persistent cargo tracking. Only the y-positions of the cargo from the trajectories are plotted for ease of comparison. Three representative moments when the cargo was detected by either TD-iSCAT only, SBR-iSCAT only, or both, marked as (i), (iii), and (ii) respectively, are captured by snapshots (left: SBR-iSCAT; right: TD-iSCAT; yellow circle: detected cargo). The trace of the cargo with SBR-iSCAT was successfully acquired except for a few moments where it was masked by an unknown object (case (i)). The cargo was frequently lost in TD-iSCAT due to its intermittent pauses (case (iii)). c, The SBR-iSCAT contrast of the cargo, reflecting its axial motion, is shown with images of the cargo in the brightest and darkest contrast next to a peak and a trough in the time trace, respectively. See also Supplementary Video 3.

SBR-iSCAT imaging. To emphasize the complementarity of both iSCAT image processing methods in persistent cargo tracking, we have added a new example in the revised manuscript, showing that the position of a cargo can be tracked continuously, or nearly so, using both SBR- and TD-iSCAT imaging simultaneously. This can be found in ‘Supplementary Fig. 3’ in the revised manuscript or Rebuttal_Figure 1.

To provide further clarification on our cargo tracking methodology using iSCAT microscopy, we have added a detailed description at the end of the section, ‘Cargos in directional motion are tracked in a complex cellular environment via iSCAT microscopy’, in the revised manuscript as follows:

(Before) In lines 19-21 on page 3, “*Thus, intracellular cargos are the main target of TD-iSCAT imaging because many native non-fluorescent cargos without containing f-PS beads are readily visualized by TD-iSCAT as indicated by orange circles (Fig. 1d)*”.

(After) “*Thus, intracellular cargos are the main target of TD-iSCAT imaging because many native non-fluorescent cargos without containing f-PS beads are readily visualized by TD-iSCAT as indicated by orange circles (Fig. 1d). Although TD-iSCAT imaging may lose track of cargos during pauses, the concurrent use of SBR- and TD-iSCAT imaging allows for continuous, or near-continuous, tracking of the cargos, as demonstrated in Supplementary Fig. 3 and Supplementary Video 3*”.

Next, we address the limitation of iSCAT in terms of chemical selectivity. The referee pointed out that iSCAT observes all particles passing through a specific region, regardless of their identity. To overcome this limitation, we integrated fluorescence microscopy into our iSCAT setup, which we refer to as F-iSCAT. The integration of fluorescence imaging provides chemical information to supplement the data acquired by iSCAT, leading to a more reliable interpretation of the results. As demonstrated in the original manuscript (Extended Data Fig. 2), TD-iSCAT was able to visualize dynamic cargos containing f-PS beads when combined with fluorescence imaging (Figs. 1d,e in the original manuscript). In the revised manuscript, we have added more data to show that F-iSCAT can track intracellular targets such as microtubules (MTs) and late endosomes (LEs) specifically, when they are fluorescently labeled (see Supplementary Figs. 10 and 12 in the revised manuscript or our point-by-point response to referee #2's comments). By doing so, we demonstrate the added value of F-iSCAT microscopy as a functional imaging tool that can study specific phenomena revealed by “all particles of a type” in the context of the cellular environment in which they occur.

Lastly, the referee noted that our original manuscript was descriptive in nature. We acknowledge this criticism that our work was conveyed in a somewhat illustrative manner. Our aim in this work was to reconstruct the intracellular traffic flow by tracking unlabeled, unidentified cargos transported along the microtubule network for an extended period in the same area of a cell. As depicted in Fig. 2 in the original manuscript, we visualized the temporal evolution of intracellular traffic flow by reconstructing approximately 10 million localization points of dynamic cargos from ~90,000 consecutive frames. To the best of our knowledge, this vast number of localization points has not been collected previously. In response to the referee's critique, we performed two additional experiments, namely the imaging of microtubules in COS-7 cells using F-iSCAT and the testing of a microtubule-targeting drug on intracellular cargo transport (presented in Supplementary Figs. 10 and 11 in the revised manuscript). These experiments were performed to enhance the statistical reliability, control, and biological significance of our results. We successfully reproduced our main results of intracellular cargo transport, confirmed the association of the mobile particles with the microtubule network as cellular cargos, and demonstrated the critical role of the network in cargo transport. We have thoroughly revised our original manuscript with these new data to support our main findings quantitatively. The details can be found in our point-by-point responses to the referee's comments

Major comments:

(Q1) It was not clear to me from the manuscript what the practical limitations are on the cargo tracking. For instance, what is the range of cargo sizes and speeds that can be robustly extracted by this method? How do the authors overcome issues of crowding with many particles crowded together within a diffraction-limited (or at least very small) zone -- would such situations prevent any cargo from being tracked? How often does that occur? Given that this method does not rely on labeling specific cargos, it seems surprising that individual particles show up at all given the many particles one would expect to be moving through any given area of the cell.

(A1) Thank you for addressing the concerns regarding the detection limit of iSCAT live-cell imaging. As noted by the referee, iSCAT imaging is subject to diffraction limit and may have difficulty in detecting single biomolecules or organelles in heterogeneous and crowded intracellular environments. Thus, one may think that this technique cannot individually identify or track cargos in crowded areas. Therefore, our iSCAT-based long-term tracking of a specific target can be occasionally or transiently interrupted for various reasons (for example, heterogeneity cast by various cellular objects, limited resolution, temporal pausing, etc.), the issue of which can be sometimes alleviated by the use of complementary fluorescence detection. As shown in Rebuttal Fig. 1 above, our iSCAT technique can track cargos almost continuously by using the two image processing methods simultaneously, even in a crowded cellular environment. As long as cargos are tracked continuously, their speeds can be readily extracted. Despite the expected difficulty, in some cases, individual cargos can be discerned even in a highly crowded area, as demonstrated in Rebuttal Figs. 3 and 4 (see our answer (A2) to Q2 below). As the referee noted, it is indeed surprising that individual particles show up after the aforementioned image analysis scheme based on the cargo's dynamics.

In the revised manuscript, we have included Supplementary Fig. 5 (Rebuttal_Figure 2) to demonstrate the size distribution of cargos captured in three SBR-iSCAT snapshots taken at $t = 0, 20,$ and 40 s. The average size of 144 cargos detected from those SBR-iSCAT snapshots was 393 ± 62 nm, as measured by the MOSAIC ImageJ plugin.

Rebuttal Fig. 3 (see **Supplementary Fig. 5** in the revised manuscript). **Size distribution of cargos in SBR-iSCAT snapshots.** a, Cargos (yellow circles) identified in a SBR-iSCAT snapshots at $t = 0$ using the MOSAIC ImageJ plugin with particle radius (r) of 7, cutoff score (S_{cut}) of 0, and intensity percentile (I_p) of 0.5. b, Histogram of the cargo size distribution for 144 cargos detected from three SBR-iSCAT snapshots taken at $t = 0, 20,$ and 40 s. The average size measured was 393 ± 62 nm.

To provide this quantitative result, we have added several sentences in the section, ‘Cargo-localization iSCAT microscopy reveals the underlying active cytoskeletal highways’, in the revised manuscript as follows:

(Before) In lines 26-38 on page 3, “*We note that cargos in directional motion are confined on cytoskeletal tracks, and the cargo location is a faithful proxy for a cytoskeletal foothold with an error of cargo size. To calculate the localization precision for a single cargo revealed in TD-iSCAT images, the pair of bright and dark spots that represents the cargo was fitted with a couple of two 2D Gaussian functions (Extended Data Fig. 3). For bright and dark spots, the value of signal-to-noise ratio (SNR), 6 and 4, allows localization of the center of their Gaussian peaks to be within 10 and 15 nm in precision, respectively. Thus, by collecting a vast number of cargo locations, one can reconstruct the underlying cytoskeletal network. We define cargo location by the center of the bright spot (Extended Data Fig. 4a) using the MOSAIC ImageJ plugin²⁹. We could visualize their underlying cytoskeletal tracks by accumulating many localization points (~ 100,000) obtained from 50,000 sequential images (Extended Data Fig. 4b). Similarly, the cytoskeletal network in lamellar and lamellipodium in a COS-7 cell was reconstructed from ~ 10-million localization points obtained from ~ 90,000 consecutive frames taken at 50 Hz (Fig. 2a)*”.

(After) “*We note that cargos in directional motion are confined on cytoskeletal tracks, and the cargo location is a faithful proxy for a cytoskeletal foothold with **the cargo size representing the error in the transverse dimension**. To calculate the localization precision for a single cargo **identified by TD-iSCAT images, its physical dimension was measured from the complementary SBR-iSCAT image**. The bright or dark spots in the SBR-iSCAT image, corresponding to the dynamic cargos observed in the TD-iSCAT image, were analyzed using 2D-Gaussian fitting as shown in Supplementary Fig. 4. The average size of a cargo measured was 393 ± 62 nm (Supplementary Fig. 5). For bright and dark spots, exhibiting signal-to-noise ratios (SNR) of 6 and 4, respectively, the centers of their Gaussian peaks can be localized with a precision of within 10 nm and 15 nm, respectively.*

By collecting a vast number of cargo locations, one can reconstruct the underlying cytoskeletal network. We define cargo location by the center of the bright spot using the MOSAIC ImageJ plugin²⁹. We visualized the cytoskeletal network in lamellar and lamellipodium in a COS-7 cell, which was reconstructed from ~ 10-million localization points obtained from ~ 90,000 consecutive frames taken at 50 Hz (Fig. 2a). The procedure of cytoskeleton reconstruction is also shown in Supplementary Fig. 6a, b”.

(Q2) Is it possible to tell the number of cargos involved in a "packet" or "cluster"?

(A2) Yes. Our study enabled us to quantify the number of cargos in both traffic jammed regions and moving packets within the diffraction limit. In the original manuscript, we reported the number of cargos continuously tracked over 500 frames (10 s) in a crowded region, as shown in Fig. 3 and related text. To provide further clarification, we have added two supplementary figures, Supplementary Figs. 8 and 13, (Rebuttal_Figs. 3 and 4) in the revised manuscript. Rebuttal_Fig. 3 shows the cargos in packets that are counted using TD-iSCAT snapshots (corresponding to three numbered red ovals (1, 2, and 3) in Fig. 2d of the revised manuscript). Rebuttal Fig. 4 displays cargos in the jammed region detected by both SBR-iSCAT and TD-iSCAT (corresponding to the white-box area in Fig. 2d of the original manuscript) and a variation graph illustrating the number of such cargos counted from both iSCAT movies.

To provide information on the number of cargos involved in a packet or a jammed intersection, we have added the following sentences in the revised manuscript:

(Before) In lines 37-38 on page 4, “At the intersection of cytoskeletons, this packet was divided into two and driven in opposite directions along the cell boundary”.

(After) “At the intersection of cytoskeletons, this packet was divided into two and driven in opposite directions along the cell boundary. From TD-iSCAT images, we could recognize individual cargos in a packet and count them (Supplementary Fig. 8)”.

(Before) In lines 14-17 on page 5, “Among them, the locations of 5 individual objects exactly matched those of moving cargos identified in the corresponding TD-iSCAT image (right in Fig. 3a). A few cargos in the TD-iSCAT image seemed to form a cluster (red circle in Fig. 3a)”.

(After) “Among them, the locations of 5 individual objects exactly matched those of moving cargos identified in the corresponding TD-iSCAT image (right in Fig. 3a). The number of cargos detected in the SBR-iSCAT images is considerably higher, in this case approximately five times greater, than those detected in the corresponding TD-iSCAT images, which exclusively display moving cargos (Supplementary Fig. 13). A few cargos in the TD-iSCAT image seemed to form a cluster (red circle in Fig. 3a)”.

Rebuttal_Fig. 3 (see **Supplementary Fig. 8** in the revised manuscript). **Cargo packets.** Densely packed clusters of cargos that move collectively, marked by red dotted ovals (1, 2, and 3) in Fig. 2d. Individual cargos within the packets are indicated by yellow arrowheads in TD-iSCAT snapshots taken at $t = 361.8$ s (a), 410.0 s (b), and 568.7 s (c). Scale bar: 1 μm .

Rebuttal_Fig. 4 (or see **Supplementary Fig. 13** in the revised manuscript). **Detection and counting of cargos in a traffic-jammed area, also shown in Fig. 3a, by SBR- and TD-iSCAT methods.** **a**, Cargos (represented as circles) in snapshots of SBR-iSCAT (left) and TD-iSCAT (right) detected by the MOSAIC ImageJ plugin. Circles in yellow represent cargos detected simultaneously in both snapshots while those in blue were only detected in SBR-iSCAT. The red-dashed circle highlights a cargo-jamming spot in the TD-iSCAT image, where at least 3 cargos (blue circles in red-dashed circle) were identified in the complementary SBR-iSCAT image. **b**, Temporal variation in the number of cargos in the area shown in (a), as detected by SBR-iSCAT (blue) and TD-iSCAT (red). Image acquisition rate: 50 Hz.

(Q3) Can hitchhiking events be observed directly, in the form of pairs of particles where one is already moving processively before picking up an extra passive hitchhiker? Or are the observed dimers only seen in their dimeric form from start to end of the trajectory?

(A3) The process of microtubule-based cargo transport involves the recruitment of molecular motors by specific adaptors for long-range transport. Recent evidence suggests that intracellular organelles can be delivered through hitchhiking on other organelles driven by motor proteins (1-4). The main focus of this question is the direct observability of hitchhiking and the stability of the dimeric form. Our literature review showed that the hitchhiking model is based on experimental observations of dimers composed

Rebuttal_Fig. 5 (or see **Supplementary Fig. 15** in the revised manuscript). **Dissociation of a dimeric cargo.** The separation of a dimeric cargo is depicted in boxed areas in SBR-iSCAT (left), TD-iSCAT (middle), and fluorescence (right) images taken at six time points. At $t \sim 15$ s, the front cargo (red box) moves forward continuously while the rear cargo (green box) stalls. The front cargo contains a f-PS bead, as indicated by fluorescence. See also Supplementary Video 11.

of a cargo and an organelle, labeled in different colors, traveling together over a long distance. No direct observations of hitchhiking via microscopic imaging have been made yet. Upon re-examination of our data sets, we observed dimeric cargos in all data sets, but most of them remained in the dimeric form for the entire observation time, typically around 30 minutes, with one exception where a dimeric cargo split into two monomeric cargos (see Rebuttal Fig. 5). The stability of the dimeric form, as indicated by the statistical information in our iSCAT experiments, suggests that the association or dissociation of the dimeric cargo is rare (see our answer to Q5 below).

In order to provide additional evidence for the existence of dimeric cargos, we have made significant revisions in the section titled ‘Dynamic phenomena revealed in cargo transport resemble commonplace events in transportation’ as follows:

(Before) *“The superb detection sensitivity of iSCAT allows various dynamic events to be revealed in the process of intracellular cargo transport. Interestingly, we found that some cargos move together as a pair, namely “dimeric cargo” (Fig. 4a). For a long observation time (> 130 s), cargos in such a dimeric form moved along the same path without being separated. However, its journey was not smooth due to many obstacles scattered along the course. The dimeric cargo moving toward the cell boundary could not move any further after colliding with a large cellular obstacle (time window ‘a’ in Fig. 4b). After stopping or randomly jiggling for a while, it changed the direction of motion and retraced its path (Supplementary Video 5). Then, its direction of motion changed once again by a head-on collision with another cargo moving oppositely along the same path (red arrow in Fig. 4b, Supplementary Video 6). The dimeric cargo stopped again at the surface of the large obstacle (time window ‘c’ in Fig. 4b), turned around, and then persistently propagated to the cell body (Supplementary Video 7).*

Because the fluorescence-combined iSCAT microscopy captures the dynamic events of cargos regardless of f-PS labeling and detects both scattered light from cargos and fluorescence from internalized f-PS beads simultaneously for precise specification, we could identify dimeric cargos straightforwardly (Extended Data Fig. 7a, Supplementary Video 8). A close inspection of the f-PS bead by iSCAT imaging revealed the f-PS bead as a part of a dimeric cargo. Interestingly, the dimeric cargo was put on hold because other cargos crossed the intersection ahead of it (Extended Data Fig. 7f). It was recently reported that intracellular cargos move together via ‘hitchhiking’. Some cargos literally can hitch a ride with nearby motile cargos already associated with molecular motors^{35,36}. Intriguingly, iSCAT can directly visualize the outcome of hitchhiking, multimeric cargo, even when some cargos lack f-PS beads inside. A cargo trimer was also observed to move together along the same route (Extended Data Fig. 8, Supplementary Video 9). During translocation, one of the cargos was detached at $t = 210.7$ s, and the rest (dimer) continued to move and turned around at the intersection of the cytoskeleton from $t = 215.7$ s to 216.8 s. The hitchhiking process could be a rule rather than an exception to boost the efficiency of logistics by implementing specific adapter proteins bridging different vesicles^{37,38}. Thus, it may be programmed for the cell’s benefit, which is somewhat analogous to carpooling.

Interestingly, the dynamic events described herein nicely match various occasions encountered during transportation: a car should detour at roadblocks, wait for other vehicles to cross at the intersection, and come to a stop at the end of the road. It is of note that a complete picture of cellular transportation is now made available by the brightfield iSCAT microscopy”.

(After) *“The superb detection sensitivity of iSCAT allows various dynamic events to be revealed in the process of intracellular cargo transport. Interestingly, we found that some cargos move together as a pair; namely “dimeric cargo” (Fig. 4a). For a long observation time (> 130 s), cargos in such a dimeric form moved along the same path without being separated. However, its journey was not smooth due to many obstacles scattered along the course. The dimeric cargo moving toward the cell boundary could not move any further after colliding with a large cellular obstacle (time window ‘a’ in Fig. 4b). After stopping or randomly jiggling for a while, it changed the direction of motion and retraced its path (Supplementary Video 8). Then, its direction of motion changed once again by a head-on collision with another cargo moving oppositely along the same path (red arrow ‘b’ in Fig. 4b, Supplementary Video 9). The dimeric cargo stopped again at the surface of the large obstacle (time window ‘c’ in Fig. 4b), turned around, and then persistently propagated to the cell body (Supplementary Video 10).*

Despite experiencing a bumpy journey, the cargo remained dimeric, suggesting that the dimeric cargos are generally stable. Within a lamellipodial region (Fig. 2c), we monitored at least eight moving dimeric cargos for ~ 30 minutes (Supplementary Fig. 14), indicating that their existence is not a rare event. Long-term tracking of dimeric cargos was challenging due to their departure from the field of view or signal interference, making it difficult to follow them to their final fate. However, their long-term stability was evident as they all remained bound throughout the entire observation period. In one exceptional case observed in another COS-7 cell, a dimeric cargo split into two monomeric cargos (Supplementary Fig. 15 and Supplementary Video 11). A cargo trimer was also observed to move together along the same route (Supplementary Fig. 16, Supplementary Video 12). During translocation, one of the cargos was detached at $t = 210.7$ s, and the rest (dimer) continued to move and turned around at the intersection of the cytoskeleton from $t = 215.7$ s to 216.8 s.

During cellular cargo transport, dynamic events reminiscent of car transportation were observed. One such example was found while monitoring a dimeric cargo. As noted, dimeric cargos are interesting in their own right and reliable, making them preferred targets for our cargo transport study. Their dumbbell shape allows for clear identification as cellular cargos, distinguishing them from other monomeric cargos in cargo-dense regions. Utilizing fluorescence-combined iSCAT microscopy, we were able to capture dynamic cargo events independent of f-PS labeling and simultaneously detect scattered light from cargos and fluorescence from internalized f-PS beads for specification. This allowed us to identify a dimeric cargo straightforwardly and accurately when associated with an f-PS bead (Supplementary Fig. 17a, Supplementary Video 13). In this instance, an f-PS bead was identified as part of a dimeric cargo upon close examination using iSCAT imaging. This dimeric cargo exhibited a sporadic journey characterized by frequent stalling, turnarounds, and unsteady movements. Interestingly, the dimeric cargo was put on hold because other cargos crossed the intersection ahead of it (Supplementary Fig. 16f). From an extensive collection of long-term tracking data, we observed that the two cargo modules in a dimeric cargo move in a highly coherent and correlated manner, suggesting that they must be physically bound. We present another compelling piece of evidence for physical binding in a dimeric cargo, which is a ‘U-turn’ during its journey (Supplementary Fig. 18). This dimer, composed of two cargo modules, was tracked for ~ 150 s. The trajectory shows that the dimer makes a U-turn within the green-boxed area (Supplementary Fig. 18a, Supplementary Video 14). In the sequence of SBR-iSCAT images showing the U-turn, the center locations of the front and rear cargos are indicated by red and blue dots, respectively (Supplementary Fig. 18b). After making a U-turn, the relative orientation of the two cargos was reversed (the front-rear relationship between the two cargos remained unchanged). It indicates that only one of the cargos (front) is actively transported by the associated

motor protein on the cytoskeleton while the other one (rear) passively trails the active tractor, hanging from it. Furthermore, the distance between the two cargos (d) measured in 2D varied but no more than ~ 350 nm, which is likely due to the 3D motion of the rear cargo (Supplementary Fig. 18c).

It was recently reported that intracellular cargos move together via 'hitchhiking'. Some cargos literally can hitch a ride with nearby motile cargos already associated with molecular motors^{38,39}. As shown above, iSCAT can directly visualize the outcome of hitchhiking, multimeric cargo, even when some cargos lack f -PS beads inside. Capturing the moment of hitchhiking is challenging, as it likely occurs at the early stage of cargo transport and near the endoplasmic reticulum (ER), where the cell's interior is too crowded with cargos and other organelles for direct visualization using iSCAT. This concentration of cargos in the ER region increases the likelihood of them coupling together. The hitchhiking process could be a rule rather than an exception to boost the efficiency of logistics by implementing specific adapter proteins bridging different vesicles^{40,41}. Thus, it may be programmed for the cell's benefit, which is somewhat analogous to carpooling.

Interestingly, the dynamic events described herein nicely match various occasions encountered during car transportation: a car should detour at roadblocks, wait for other vehicles to cross at the intersection, and come to a stop at the end of the road. It is of note that a complete picture of cellular transportation is now made available by the brightfield iSCAT microscopy”.

[References]

1. Salogiannis, J. & Reck-Peterson, S. L. Hitchhiking: A non-canonical mode of microtubule-based transport. *Trends Cell Biol.* **27**, 141-150 (2017) (cited as 36 in the original manuscript).
2. Mogre, S. S., Christensen, J. R., Niman, C. S., Reck-Peterson, S. L. & Koslover, E. F. Hitching a ride: Mechanics of transport initiation through linker-mediate hitchhiking. *Biophys. J.* **118**, 1357-1369 (2020) (cited as 37 in the original manuscript).
3. Salogiannis, J., Egan, M. J. & Reck-Peterson, S. L. Peroxisomes move by hitchhiking on early endosomes using the novel linker protein PxdA. *J. Cell Biol.* **212**, 289-296 (2016) (cited as 38 in the original manuscript).
4. Christensen, J. R. & Reck-Peterson, S. L. Hitchhiking across kingdoms: cotransport of cargos in fungal, animal, and plant cells. *Annu. Rev. Cell Dev. Biol.* **38**, 155-178 (2022).

(Q4) Can statistics on microtubule distribution be extracted from this data. For example, the authors mention a persistence length of 4-8mm for MTs -- the actual bending in vivo has been measured to be much greater, due to active fluctuations (see, eg, Brangwynne, JCB, 2008). Can a persistence length be extracted from the cargo trajectories? How about the location or temporal persistence of MT intersections?

(A4) We appreciate the referee's insightful suggestion regarding the potential to extract statistical information about microtubules from our data. Just as with super-resolution fluorescence microscopy, the CL-iSCAT microscopy technique we employed enables us to visualize the overall structural layout of the microtubule network, as well as the locations of MT intersections (see Supplementary Fig. 10 in the revised manuscript), with high spatial resolution. Furthermore, it allows us to monitor the actively-used MT network over an extended period of time.

Having carefully reviewed the work by Brangwynne et al. (5), we recognize the importance of directly observing temporal fluctuations of individual microtubules in order to accurately calculate their

persistence length. Unfortunately, our current CL-iSCAT methodology does not allow for the tracking of individual microtubule fluctuations. Rather, it reconstructs the spatial organization of the microtubule network from the localization points of cargos accumulated over a much longer time than the time scale characteristic of microtubule filament bending.

Due to the low acquisition rate in our experiment, we are currently unable to directly measure the elastic properties of microtubules. Nevertheless, we acknowledge the potential of high-frequency cargo tracking in extracting the persistence length of cargo-bound microtubules. By reducing the scanning area and increasing the imaging speed, it may be possible to further explore this aspect in future research. This promising avenue could substantially expand the biological applications of CL-iSCAT microscopy and enhance our understanding of microtubule dynamics.

[References]

5. Brangwynne, C. P. et al. Cytoplasmic diffusion: molecular motors mix it up. *J. Cell Biol.* **183**, 583-587 (2008).

(Q5) It was not clear to me how hitchhiking (in the sense of a direct molecular connection between cargos) could be distinguished from two particles walking close to each other (within the diffraction resolution limit) with similar velocities. Perhaps statistics on how long cargo dimers and trimers persist could be collected? Or statistics on the distribution of packet sizes (how many particles are moving alone vs in larger groups)? These are just suggestions, and not specifically necessary for the paper. But it does seem that the paper would be improved by some quantitative results beyond the velocity measurements which have been previously carried out.

(A5) We are grateful for the referee's suggestions, which could indeed enhance the rigor and impact of our manuscript. As mentioned earlier, our evidence for hitchhiking events is based on the correlated and coordinated motion of two nearby particles traveling together for extended distances (please refer to our response to Q3). To substantiate this claim, we have provided the following physical data:

(i) U-turn of a dimeric cargo: The characteristic feature of hitchhiking would be that only one of the cargos (or tractor cargo) is actively transported by the associated motor protein on cytoskeleton while the other one (or trailer cargo) passively moves with the active tractor, hanging from it. In the revised manuscript, we provided one such example, which exhibits the event of 'U-turn' during a journey (Rebuttal_Fig. 6). This dimeric cargo was tracked over a long distance ($\sim 10 \mu\text{m}$) for ~ 150 s. As shown in the trajectory (Rebuttal_Fig. 6a), the dimer makes a U-turn in the green boxed area. In the sequence of SBR-iSCAT images showing a U-turn, the center locations of the front and rear cargos are indicated by red and blue dots, respectively (Rebuttal_Fig. 6b). After making a U-turn, the relative orientation of the two cargos was reversed (the front-rear relationship between the two cargos remained unchanged) as indicated by the red and blue arrows, showing the motion of the front and rear cargos, respectively. Furthermore, the distance between the two cargos (d) measured in 2D varied but no more than ~ 350 nm, which is likely due to a 3D motion of the trailer (Rebuttal_Fig. 6c). We are confident that the observed coordinated motion of front and rear cargos provides compelling evidence for a direct molecular connection between these cargos.

(ii) Statistics of dimeric cargos traveling together: Our observations showed that dimeric cargos are frequent. From one area of a lamellipodium (Fig. 2c in the original manuscript), we observed at least eight dimeric cargos traveling together for about 30 minutes (Rebuttal_Fig. 7). While long-term

tracking of cargos was challenging due to their departure from the field of view or signal interference, we were able to successfully track eight dimeric cargos, as depicted in Rebuttal_Fig. 7a using distinct colors. Throughout the time intervals indicated in parentheses in Rebuttal_Fig. 7b, the cargos maintained their dimeric state, indicating their stability over a long journey.

Rebuttal_Fig. 6 (or see **Supplementary Fig. 18** in the revised manuscript). U-turn of a dimeric cargo. a, Trajectories of the two cargos moving in a dimeric form (front cargo: color-coded by elapsed time (see color legend); rear cargo: gray). Insets: Snapshots of the dimeric cargo taken at $t = 25.74$ s using SBR-iSCAT (upper) and TD-iSCAT (lower) imaging. b, Sequence of images taken at five time points showing the U-turn of the dimeric cargo (red dot: front cargo; blue dot: rear cargo) and the zoomed-in trajectories in the green-boxed area in (a). Color and grayscale legends represent time points for front and rear cargos, respectively. The direction of motion of the front and rear cargos is indicated by red and blue arrows, respectively. c, The distance between the front and rear cargos remained less than 300 nm during observation. The center location of each cargo was calculated by fitting with a 2D Gaussian function. See also Supplementary Video. 14

Rebuttal_Fig. 7 (or see **Supplementary Fig. 14** in the revised manuscript). **Long journey of dimeric cargos observed by iSCAT microscopy.** (a) Trajectories of eight dimeric cargos tracked within a single lamellipodial region, as shown in Fig. 1f. The dotted line indicates the cell boundary. (b) Snapshots of the eight dimeric cargos captured by TD-iSCAT imaging. The colored number in parenthesis indicates the period of the successful cargo tracking for each dimer. The same color is used to represent the same dimeric cargo in both (a) and (b). The insets in each panel of (b) show the corresponding SBR-iSCAT images, which depict the shape of the dimeric cargo.

(Q6) Along the same lines, the generalized claims about how cargos move collectively on longer time-scales seem a bit overstated, as only one or two example cases of such motion are shown. How often does this occur? Is this a general feature or an outlier?

(A6) We understand the referee's concern that example cases presented in the original manuscript are not sufficient to support our generalized claim. In the original manuscript, we proposed that cells have efficient strategies to overcome traffic problems such as forming dimeric cargos or moving cargos collectively in packets. In the revised manuscript, we have included many more examples of dimeric cargos and cargos traveling together (as mentioned in our answers (A3 and A5) above). We also noted that similar cases were reported in recent studies using fluorescence-based imaging techniques, as explained in lines 12-14 on page 6 of the original manuscript (lines 7-15 on page 8 of the revised manuscript). Although to unveil the detailed mechanisms underlying the hitchhiking model requires further investigation, the presence of dimeric cargos and collective motion of cargos seem to be general features of cellular transport.

Minor comments:

(Q7) Line 110-111: the MT polarities seem to be reversed. (+) ends should be towards the cell boundary.

(A7) Thank you for pointing out the error. We corrected it. We have also corrected additional directionality-related errors that were not previously identified. These errors were present in the descriptions of Supplementary Figure 4c, d in the original manuscript (Supplementary Fig. 6c, d in the revised manuscript) as follows.

(Before) In lines 20-30 on page 4, “Here, we selected 9 and 10 cargos moving to the (+) and (–) ends, respectively, continuously tracked over at least 3 s (> 150 consecutive frames) with iSCAT. Before evaluating the ‘instantaneous’ speed of cargo, we first acquired the ‘moving-averaged’ position to suppress noise in speed measurement. The x and y positions of cargo at 50 consecutive time points in a raw trajectory were averaged to yield the position of the cargo in the 50-point time-averaged (1 s) trajectory (Extended Data Fig. 5). Then, we calculated the instantaneous speed from the displacement made by the cargo for 0.02 s and collected such data to get the distributions of instantaneous speeds of cargos moving to the (+) and (–) ends (Extended Data Fig. 4e). From the distributions, the average speeds of cargos moving to the (+) and (–) ends were $0.53 \pm 0.28 \mu\text{m/s}$ and $0.45 \pm 0.19 \mu\text{m/s}$, respectively. Kinesin and dynein motors are the key players responsible for transporting cargos toward the (+) and (–) ends of a microtubule, respectively³²”.

(After) “Here, we selected 9 and 10 cargos moving to the (–) and (+) ends, respectively, continuously tracked over at least 3 s (> 150 consecutive frames) with iSCAT (Supplementary Fig. 6c, d, respectively). Before evaluating the ‘instantaneous’ speed of cargo, we first acquired the ‘moving-averaged’ position to suppress noise in speed measurement. The x and y positions of cargo at 50 consecutive time points in a raw trajectory were averaged to yield the position of the cargo in the 50-point time-averaged (1 s) trajectory (Supplementary Fig. 7 and Supplementary Video 5). Then, we calculated the instantaneous speed from the displacement made by the cargo for 0.02 s and collected such data to get the distributions of instantaneous speeds of cargos moving to the (–) and (+) ends (Supplementary Fig. 6e). From the distributions, the average speeds of cargos moving to the (–) and (+) ends were $0.45 \pm 0.19 \mu\text{m/s}$ and

$0.53 \pm 0.28 \mu\text{m/s}$, respectively. *Dynein and kinesin motors are the key players responsible for transporting cargos toward the (-) and (+) ends of a microtubule, respectively³²*”.

(Q8) Line 113-114: vesicles depart from the ER in the cell periphery as well. Also many cargos presumably depart from the Golgi and from the plasma membrane following the endocytic pathway.

(A8) Our apologies for the confusion. We did mention in the first paragraph of the original manuscript “*For cargo transport, vesicles are constantly formed as a container at the cell membrane, endoplasmic reticulum (ER), and Golgi apparatus^{3,4} and transported by motor protein-driven activity on cytoskeletal highways*”. However, in line 113-114, our focus was solely on why the cargo density is higher in the cell body compared to peripheral regions, as shown in Figure 2c in the original manuscript. Given the close proximity of the ER to the nucleus and its sparsity outward, we still believe that our description in line 113-114 of the original manuscript is accurate.

(Q9) Figure 1c: the dark blue oval is nearly invisible in print

(A9) In the revised manuscript, we changed it to a white-dotted oval.

(Q10) Figure 2c: It is not clear what is meant by "total number of cargos detected at each pixel for 90000 consecutive frames". Is this distinct cargos? Are does the same cargo found in one pixel for multiple frames (either from slow movement or because it backtracks) get counted multiple times?

(A10) We appreciate the referee's inquiry. The description in the original manuscript was not entirely clear. In our SBR-iSCAT imaging system, if a cargo remains in a fixed location (or pixel), it is counted multiple times. To address this issue, we created the traffic density map (Fig. 2c) from only dynamic cargos detected by TD-iSCAT images. The time interval was set to 0.2 s and the pixel size was 40 nm, which means cargos that move slower than ~ 200 nm/s (including those in pause) are not counted as they disappear in TD-iSCAT images. According to literature, motor proteins move at a speed of ~ 200 nm/s or faster, up to a few $\mu\text{m/s}$. The referee correctly pointed out that cargos that retrace their routes would be counted multiple times. In the revised manuscript, we added the following clarification:

(Before) In lines 28-31 of the subsection, ‘TD-iSCAT image processing’ in ‘Methods’, “*In the image, individual cargos appear as a pair of dark and bright spots, indicating the early and late positions in the time interval averaged over for each image ($\Delta t = 0.4$ s). In contrast, other static or randomly moving cellular entities fade or disappear due to signal cancellation*”.

(After) “*In the image, individual cargos which move faster than 200 nm/s appear as a pair of dark and bright spots, indicating the early and late positions in the time interval averaged over for each image ($\Delta t = 0.4$ s). In contrast, other static or randomly moving cellular entities fade or disappear due to signal cancellation. Cargos slower than 200 nm/s would be self-canceled because their signals fall in the same pixel (1 pixel ~ 40 nm) within the time interval of 0.2 s. In the cases cargos backtrack, however, they would be counted multiple times*”.

(Q11) Figure 4: Can the authors comment on the identity of the large cargos observed (which look beyond diffraction-limited in size)? Are these also vesicles containing fPS beads or are these unrelated particles?

(A11) We appreciate the referee's interesting question regarding the identity of the 'large cargos' observed in our study, which appear to be beyond the diffraction-limited size. We concur that iSCAT imaging reveals a variety of cytoplasmic organelles, some of which are larger than the diffraction limit. However, determining the precise identity of these structures using iSCAT alone is not feasible without supplementary information from fluorescence or chemical imaging techniques.

In our original manuscript, we cautiously referred to these large structures as "large spherical cytoplasmic structures" in the caption of Figure 4 to avoid any potential misinterpretation. Importantly, our F-iSCAT observations provided valuable insights into the nature of these structures. Throughout the entire observation period of approximately 30 minutes, these structures remained stationary and did not emit any fluorescence signals. This lack of fluorescence indicates that they do not contain f-PS beads and are therefore unrelated to the vesicles under investigation in our study.

Referee #2

(Overall remark) The work by Park et al. titled ‘Long-term cargo tracking reveals intricate trafficking through active cytoskeletal networks in the crowded cellular environment’ describes the use of iSCAT microscopy to follow intracellular trafficking of cargo vesicles along cytoskeletal filaments. The authors study Cos-7 cells and use internalized 20 nm fluorescent beads as an additional marker for vesicle cargo transport using fluorescence microscopy in parallel to iSCAT.

The motivation of this work is that iSCAT microscopy offers the benefit of label-free microscopy without the risk of photobleaching, and hence the possibility to acquire long timelapse movies at high imaging frequencies. This enables to follow intracellular dynamics of light-scattering organelles without any selectivity that occurs in typical experimental setting relying on specific labelled proteins. The authors demonstrate successfully and convincingly that this approach is indeed applicable to follow transport of vesicular cargo along cytoskeletal filaments within Cos-7 cells. Recording the edges of cells and performing two different ways of post imaging processing, i.e. static background removal (SBR) and time-differential (TD), the authors reveal a multitude of cargo shapes and dynamic behaviors. To quantify cargo speeds and trajectories, particle tracking with the MOSAIC toolbox via image J is employed.

Though this work presents an interesting approach to visualize cargo transport within live cells, it is purely descriptive and lacks any controls to support the claims made by the authors about the reasons of the observed dynamics. The number of cells imaged and analyzed is low and deep analysis of interesting cargo transport tracks seems very anecdotal lacking any wider analysis about e.g. the relative occurrence of the different observed trafficking behaviors. Another aspect that could have been investigated to provide some further insights into cellular trafficking and could have highlighted the potential of the presented methodology would be to use more sophisticated analysis of the particle tracks. The high temporal resolution would allow to distinguish between phases of diffusive and directional transport and how this relates e.g. to the presence of other cargo vesicles in the surrounding. Very important or rather essential control experiments are missing, such as using fluorescent markers for specific cellular endosomes (e.g. Rab-5, Rab-7 or Lamp-1) or a check whether the cargos detected are preferentially linked to microtubules or actin filaments. Another set of experiments that could underline that the observed phenomena are related to active transport of vesicles would be to use chemical inhibitors (e.g. blebbistatin, actin or microtubule depolymerizing drugs). Without such control experiments, it remains unclear whether all types of cargo are visualized with iSCAT or whether it is a particular subset. The time window used in the TD analysis could bias the detection for only those cargoes that move fast enough, and it would be good if the authors could comment on that.

Taken together, the work in its current form is not suitable for publication in Nature Communications. Below are further minor and technical comments.

(Reply) We would like to express our gratitude to the referee for raising several important points and challenging issues, which we believe have improved the impact and rigor of our work. As we understand it, the referee pointed out that our main findings were presented descriptively using a limited number of cases without control experiments. In the original manuscript, we primarily focused on highlighting the unique capabilities of CL-iSCAT microscopy in visualizing intracellular traffic flows in specific regions of a cell. To the best of our knowledge, this type of functionality or level of performance has not been achieved by other fluorescence or label-free imaging methods. As a result, instead of presenting multiple examples from different cells, we chose to emphasize specific features from a single

cell. In response to the referee's criticisms, we have conducted additional experiments that provide robust support for our main findings, as described below.

(i) Investigating intracellular traffic flow along microtubules: Cells have a well-organized microtubule-based transport system that enables the efficient transport of organelles and vesicles over long distances, powered by kinesin and dynein motor proteins. In the revised manuscript, we provide evidence that intracellular traffic phenomena are a common feature related to microtubule-based cellular transports. To confirm the idea, we imaged a COS-7 cell expressing GFP-labeled microtubules using F-iSCAT microscopy (as shown in Rebuttal_Fig. 8 or Supplementary Fig. 10 in the revised manuscript). By comparing the cargo-localization density map (Rebuttal_Fig. 8e) with the fluorescence image (Rebuttal_Fig. 8d), we observed a clear correlation between the two. For instance, intersections of microtubules (represented by blue circles in Rebuttal_Fig. 8d) were co-localized with areas of high cargo density in the cargo-localization density map (represented by white circles in Rebuttal_Fig. 8e). From analyzing the trajectories of individual cargos superimposed on the TD-iSCAT snapshot (Rebuttal_Fig. 8f), we found that the microtubule pathways connecting these intersections were frequently used as primary transport routes.

Rebuttal_Fig. 8 (or see **Supplementary Fig. 10** in the revised manuscript). **Dual-channel F-iSCAT imaging captures the intracellular transport network of a COS-7 cell.** **a**, Fluorescence image displaying the spatial distribution of microtubules labeled with GFP. **b**, SBR-iSCAT and **c**, TD-iSCAT snapshots corresponding to the fluorescence image. Grayscale legends in (b) and (c) are used as in Fig. 1c and 1d, respectively. **d**, Magnified fluorescence image of the red-boxed area in (a). This area highlights seven microtubule intersections, marked by blue circles. **e**, The cargo-localization density map was created from 15,000 consecutive iSCAT images taken at 50 Hz, showing microtubule intersections as concentrated areas of cargo (white circles). Here, color legend is used as in Fig. 2c except for using 15,000 frames ($\Delta T = 300$ s). **f**, 402 trajectories of individual cargos superimposed onto the magnified TD-iSCAT image of the red-boxed area in (c) displays the active transport routes through the microtubule network. The white circles in (e) and (f) that correspond to the microtubule intersections marked by blue circles in (d) also exhibit high density regions in the density map in (e) and serve as primary passageways for cargo transportation where multiple cargo trajectories intersect with each other as shown in (f). The cargos selected here are among those tracked continuously over 1 s.

(ii) Drug-inhibition test to induce the change in intracellular traffic flows: To demonstrate that the vesicle transport observed through CL-iSCAT microscopy is associated with microtubule networks, we conducted a drug-inhibition test using nocodazole. Previous studies have shown that the removal or stabilization of microtubules through drugs can significantly decrease microtubule-mediated vesicle transport (6-8). Our results showed that after treatment with 2 μM nocodazole, an anticancer medication that reversibly disrupts the formation of microtubules, there was a significant reduction in traffic flows compared to the levels observed before treatment, indicating the important role of microtubules in vesicle transport (Rebuttal_Fig. 9).

Rebuttal_Fig. 9 (or see **Supplementary Fig. 11** in the revised manuscript). **Impact of nocodazole treatment on intracellular traffic in a COS-7 cell.** **a**, SBR-iSCAT snapshots of a COS-7 cell before and **b**, after treatment with 2 μM nocodazole. Nocodazole was treated at $t = 20$ min. Grayscale legend is used as in Fig. 1d. **c**, Locations of dynamic cargos (orange circles) identified from TD-iSCAT snapshots (yellow-boxed area in (a, b)) taken at two different time point, $t = 0$ and 28 min. **d**, Sequential image of cargo localization density maps of the yellow-boxed area in (a, b). Each image was generated from 9,000 consecutive frames taken at 50 Hz. Supplementary Video 5 and 6 show cargo dynamics before and after nocodazole treatment, respectively. In (d), color legend is used as in Fig. 2d. Scale bars: 2 μm .

[References]

- Breitfeld, P. P., McKinnon, W. C. & Mostov, K. E. Effects of nocodazole on vesicular traffic to the apical and basolateral surfaces of polarized MDCK cells. *J. Cell Biol.* **111**, 2365-2373 (1990) (now cited as (35) in the revised manuscript).

7. Hamm-Alvarez, S. F. & Sheetz, M. P. Microtubule-dependent vesicle transport: Modulation of channel and transporter activity in liver and kidney. *Physiol. Rev.* **78**, 1109-1129 (1998) (now cited as (36) in the revised manuscript).

8. Chenouard, N., Xuan, F. & Tsien, R. W. Synaptic vesicle traffic is supported by transient actin filaments and regulated by PKA and NO. *Nat. Commun.* **11**, 5318 (2020) (now cited as (37) in the revised manuscript).

(iii) Simultaneous tracking of specifically labeled organelles with unspecified cargos using F-iSCAT microscopy: Using F-iSCAT microscopy, we simultaneously tracked specifically labeled organelles with unknown cargos. We visualized GFP-labeled late endosomes (LEs) by fluorescence signal in COS-7 cells. We have identified 14 LEs and tracked their trajectories, as demonstrated by the numbered yellow circles and corresponding colored lines in Rebuttal_Fig. 10a and 10d, respectively. However, some cargos observed in the TD-iSCAT image were not fluorescent, indicating they were not LEs (as shown in red circles in Rebuttal_Fig. 10c). In this particular case, LEs amount to one-third of

Rebuttal_Fig. 10 (or see **Supplementary Fig. 12** in the revised manuscript). **Imaging of GFP-labeled late endosomes (LEs) in a COS-7 cell using F-iSCAT microscopy.** **a**, Fluorescence image showing 14 LEs marked by numbered yellow circles. **b**, SBR-iSCAT and **c**, TD-iSCAT images corresponding to (a). Yellow circles with the same numbers in (a) indicate co-localized cargos, *i.e.*, LEs detected in TD-iSCAT image, while red circles indicate other cargos only detected in TD-iSCAT image. Grayscale legends in (b) and (c) are used as in Fig. 1c and 1d, respectively. **d**, Trajectories of 14 LEs numbered as in (a) and tracked at 10 Hz using fluorescence microscopy, depicted in different colors. (Inset: typical snapshots of LE (#4) captured by fluorescence (F), SBR-iSCAT and TD-iSCAT images.) **e**, Cytoskeleton network reconstructed from TD-iSCAT images using cargo localization, based on 15,000 consecutive frames taken at 50 Hz. Four representative trajectories of LEs (#2, #3, #4, and #14) from (d) are overlaid. **f**, Traffic density map showing the total number of cargos detected at each pixel for a total of 15,000 consecutive frames ($\Delta T = 300$ s), indicated by the color legend. See also Supplementary Video 3.

the dynamic cargos detected by TD-iSCAT in this region. The trajectories of the LEs were well-overlapped with dense regions of the traffic density map constructed from the entire 15,000-frame image sequence, taken over 300 seconds. For instance, the 4 representative trajectories of the LEs were

well-aligned with the cytoskeletal network revealed by CL-iSCAT (as shown in Rebuttal Fig. 10e and 10f).

To describe this new result, we have added two paragraphs in the revised manuscript as follows:
(Newly added paragraph) In the 5th paragraph in the section, ‘Cargo-localization iSCAT microscopy reveals the underlying active cytoskeletal highways’ in the revised manuscript, *“To support that intracellular traffic phenomena are primarily microtubule-based cellular events, we imaged a COS-7 cell expressing GFP-labeled microtubules using F-iSCAT microscopy (Supplementary Fig. 10). The cargo-localization density map (Supplementary Fig. 10e) exhibited a remarkable correlation with the fluorescence image (Supplementary Fig. 10d) obtained simultaneously. For instance, intersections of microtubules (circles in Supplementary Fig. 10d) were co-localized with areas of high cargo density in the density map (Supplementary Fig. 10e) and common passageways for cargo transport (intersections of cargo trajectories) (Supplementary Fig. 10f). Further analysis of the trajectories of individual cargos superimposed on the TD-iSCAT snapshot (Supplementary Fig. 10f) revealed that the microtubule filaments connecting these intersections were frequently used as primary transport routes. When microtubule polymerization was blocked by treatment with 2 μ M nocodazole, the traffic flows were significantly reduced compared to the levels observed prior to treatment. (Supplementary Fig. 11 and Supplementary Video 6 and 7). This observation is consistent with previous studies illustrating that the removal or stabilization of microtubules through drugs can significantly decrease microtubule-mediated vesicle transport.³⁵⁻³⁷*

Furthermore, we succeeded in uncovering the identity of some cargos by concurrently tracking them in both iSCAT and fluorescence channels. For example, we fluorescently labeled late endosomes (LEs) in a COS-7 cell and identified 14 dynamically transported LEs, among others (yellow circles in Supplementary Fig. 12a, d). The trajectories of the LEs were well-overlapped with dense regions of the traffic density map reconstructed from the entire 15,000-frame image sequence, taken over 300 seconds. Four representative trajectories of the LEs were well-aligned with the cytoskeletal network revealed by CL-iSCAT (Supplementary Figs. 12e and f). See also Supplementary Video 3”.

We have also provided detailed procedures for control experiments in the section, ‘Cell culture procedures’, in ‘Methods’ in the revised manuscript as follows:

(Before) In lines 11-15 on page 18, *“Thus, directional cargos containing f-PS beads are likely intracellular vesicles. Then, the sample dish was loaded into a mini-incubating chamber (Chamlide, Live Cell Instrument, Korea) mounted on the piezo stage (MZS500, Thorlabs, USA) to maintain the temperature and pH in the sample dish during a long-term iSCAT live-cell imaging”.*

(After) In lines 11-15 on page 18, *“Thus, directional cargos containing f-PS beads are likely intracellular vesicles. For live-cell fluorescence imaging, 2 μ l of BacMam 2.0 reagent per 10,000 cells were added to COS-7 cells in media. We used two different BacMam 2.0 reagents to label tubulin (CI0611, Invitrogen, USA) and LEs (CI0588, Invitrogen, USA). Cell samples were incubated for at least 16 hours at 37°C. For (F-)iSCAT imaging, the sample dish was loaded into a mini-incubating chamber (Chamlide, Live Cell Instrument, Korea) mounted on the piezo stage (MZS500, Thorlabs, USA) to maintain the temperature and pH of the dish during a long-term iSCAT live-cell imaging”.*

Minor comments:

(Q1) Page 2, line 6: ‘paralleling’: wording should be changed here. I think, the authors mean something like ‘similar to’ or ‘emulating’

(A1) In response to the referee's suggestion, we revised the wording in the manuscript accordingly.

(Before) “During this transport, cargos exhibit rich and intricate dynamic events such as directional movement, intermittent pausing, turn-around, and road change at the intersection of cytoskeletal highways, paralleling manufactured vehicles operating in the urban road network (Fig. 1a)”.

(After) “During this transport, cargos exhibit rich and intricate dynamic events such as directional movement, intermittent pausing, turn-around, and road change at the intersection of cytoskeletal highways, *similar to* manufactured vehicles operating in the urban road network (Fig. 1a)”.

(Q2) Page 2, line 7: ‘unclear how these cargoes overcome...’: the traffic analogy might be quite appealing at first sight, but I wouldn’t take it too literally.

(A2) We acknowledge the referee's concerns regarding our use of a traffic analogy to explain intracellular transport. We are aware that this approach is not new and has been previously utilized in other studies to facilitate the intuitive understanding of cellular transport. Despite this, we believe the comparison is useful and have included references (9,10) to support this viewpoint. Therefore, we have decided to maintain the analogical representation in our work.

[References]

9. Liberali, P. Snijder, B. & Pelkmans, L. A hierarchical map of regulatory genetic interactions in membrane trafficking. *Cell*. **157**, 1473-1487 (2014). Also, see the news article published as “Cellular traffic control system mapped for the first time” in *Phys.org/news/2014-06-cellular-traffic.html*, provided by Univ. Zurich.

10. Leduc, C. et al. Molecular crowding creates traffic jams of kinesin motors on microtubules. *Proc. Natl. Acad. Sci. U. S. A.* **109**, 6100-6105 (2012). Also, see the commentary written by Ross, J. L. The impacts of molecular motor traffic jams, *Proc. Natl. Acad. Sci. U. S. A.* **109**, 5911-5912 (2012).

(Q3) Page 2, line 10: It's not clear how cell metabolism comes in here. Metabolism, i.e. the conversion of nutrients into energy storage and finally use of the energy to perform tasks is usually linked to ATP and mitochondria activity, not with cargo transport.

(A3) In response to the referee's criticism, we revised the wording in the manuscript as below.

(Before) “Our current understanding of cell metabolism has been brought about by fluorescence-based imaging techniques”.

(After) “Our current understanding of *cellular transport* has been brought about by fluorescence-based imaging techniques”.

(Q4) Page 2, line 25: localization data (), The authors should clarify what type of data this is. Tracks? Or the individual localization during the entire course of the time lapse?

(A4) In response to the referee's inquiry, we have revised the terminology from "localization data" to "localization points" and included information on the time required for acquiring the specified number of localization points in the updated manuscript.

(Before) *“The enormous amount of cargo localization data ($> 10^8$) enables us to reconstruct the fine architecture of cytoskeletal meshwork and visualize the temporal evolution in traffic flow along the active cytoskeletal highways”.*

(After) *“The enormous amount of cargo localization points ($> 10^8$) **acquired at 50 Hz throughout the entire observation time (~ 30 minutes)** enables us to reconstruct the fine architecture of cytoskeletal meshwork and visualize the temporal evolution in traffic flow along the active cytoskeletal highways”.*

(Q5) Page 2, line 36: feeding of Cos-7 cells: some more detail might be of help here describing how beads were internalized into cells. Do you know via which pathway they get internalized? Do they end up in membrane bound vesicles?

(A5) We apologize for the insufficient detail provided in the original manuscript concerning the internalization of beads into Cos-7 cells. It is well known that cells can take up extracellular particles through two distinct processes, phagocytosis and pinocytosis (11). While phagocytosis is primarily found in immune cells, pinocytosis occurs in all types of cells and can be divided into several distinct mechanisms, including macro-pinocytosis, clathrin-mediated endocytosis, caveolae-mediated endocytosis, and clathrin- and caveolae-independent endocytosis. Although our experiments have shown that the internalized f-PS beads are transported in vesicles based on their scattering and fluorescence signals, we were not able to determine by what specific mechanism f-PS beads are internalized by COS-7 cells. Thank you for bringing this to our attention.

[Reference]

11. Conner, S. D. & Schmid, S. L. Regulated portals of entry into the cell. *Nature*. **422**, 37-44 (2003).

(Q6) Page 3, line2-3: Overly complicated wording. Use active language and simply say what you did.

(A6) In the response to the referee's comment, we modified this sentence in the revised manuscript as follows:

(Before) From the line 41 on page 2 to the line 5 on page 3, *“Filopodia are also clearly shown at the cell edge (white arrows; Fig 1c). Nanoscale cargos may still be challenging to detect in the presence of complicated cytoplasmic background. To separately detect the scattering signals of nanoscale cargos in the cytoplasmic background, we should resort to the dynamic nature of cargo. One remarkable feature in cellular cargo transport is (bi-)directionality with intermittent pauses, as in the case of motor vehicles on busy two-way roads”.*

(After) *“Filopodia are also clearly observed at the cell periphery (white arrows; Fig. 1c). **However, identifying nanoscale cargos amid the complex cytoplasmic environment poses a significant challenge. To selectively detect the scattering signals of the cargo, we leveraged the dynamic properties inherent***

to the cargo's motion. A notable characteristic of cellular cargo transport is its (bi-)directionality, punctuated by intermittent pauses”.

(Q7) Page 3, line 4-5: Please remove the motor vehicle analogy. It is unnecessary here.

(A7) In the response to the referee’s comment, we removed this phrase in the revised manuscript.

(Before) *“One remarkable feature in cellular cargo transport is (bi-)directionality with intermittent pauses, as in the case of motor vehicles on busy two-way roads”.*

(After) *“One remarkable feature in cellular cargo transport is (bi-)directionality with intermittent pauses”.*

(Q8) Page 4, line 17: TD-iSCAT is not a tracking routine. So how should it lose track? Clarify the wording, please.

(A8) We apologize for any confusion regarding the statement about TD-iSCAT losing track of cargos. To track a cargo, it is necessary not to lose its position at any time. TD-iSCAT is an image processing method used to capture the position of dynamic cargos in the complex cytoplasmic landscape from iSCAT image data. As a temporal-median cell image is divided by another temporal median image taken 0.2 seconds later in our TD-iSCAT scheme, instantaneously paused cargos may disappear due to signal cancellation. This is why in the original manuscript, we stated that TD-iSCAT *“frequently loses track of cargos whenever they intermittently stop (Supplementary Fig. 2d).”* Thank you for bringing this to our attention.

(Q9) Page 5, line 1-2: ‘funneled into the building’: This is nice, colorful language but maybe not the most suitable for the results section of a scientific article. In addition, the observations here seem very anecdotal, and it is difficult to see what the bigger message is here.

(A9) We appreciate the referee's feedback on our description of Fig. 2d and Extended Data Fig. 6b in the original manuscript. We acknowledge that our use of the term “funneled into the building site” was too figurative for a scientific article. In the revised manuscript, we have rephrased the related sentences to more accurately convey our findings. Our goal was to demonstrate how we gained physical insight into the coordination and organization of intracellular cargo transportation by observing it at a large scale and over a long period of time.

(Before) In line 1-2 on page 5, *“A growing filopodium viewed as a waterfront construction site, our observation implies that packets of cargos were funneled into the building site and terminated for unloading (Extended Data Fig. 6)”.*

(After) *“We also observed that cargo packets converged toward the point from which a filopodium is growing (Supplementary Fig. 9)”.*

(Q10) Page 5, line 8-9: It is not clear what the authors want to describe here. What is meant by backdrop? Do you want to say that iSCAT has the benefit of providing both, information about the overall organisation/ build-up of the cell (e.g. by taking the average of the timelapse) and about different dynamic processes (by analysing the differential imaging information)?

(A10) Yes, that is correct. The term "backdrop" in Fig. 1c refers to the contrast-enhanced static subcellular landscape acquired by applying the SBR-iSCAT image processing method. This method reveals various subcellular structures by enhancing image contrast while the TD-iSCAT method captures dynamic features, such as moving cargos, by suppressing scattering signals from relatively static subcellular structures. We apologize for any confusion and have revised the original manuscript to provide a clearer explanation. The relevant information can be found in lines 32-37 on page 2.

(Before) In lines 8-9 on page 5, *“One of the advantages of iSCAT for live-cell imaging is its capability of providing the brightfield image of the entire area and simultaneously visualizing the backdrop as well as an event of interest”*.

(After) In lines 8-10 on page 5, *“One of the advantages of iSCAT for live-cell imaging is its capability of providing the brightfield image of the entire cellular area, visualizing dynamic events of interest as well as the static subcellular background by applying SBR- and TD-iSCAT image processing methods”*.

(Q11) Page 5, line 18, use ‘unsuitable’ instead of ‘unusable’

(A11) According to the referee’s comment, we corrected it in the revised manuscript.

(Q12) Page 6, line 39-40: It is true that the iSCAT localisations mainly reflects the effective cytoskeletal transport paths, however, tracks that are remodelled, i.e. which change along the time of taking, will not be represented well.

(A12) We are grateful for the referee's recognition of the unique aspect of our work, where CL-iSCAT primarily uncovers the effective cytoskeletal transport paths. We understand and appreciate the referee's concern about the limitation of our current CL-iSCAT imaging method in capturing remodeled tracks.

In response to this concern, we acknowledge that the existing approach relies on the accumulation of cargo localization points from thousands of frames, which span a significantly longer time than the time scale needed for observing microtubule remodeling processes. As a result, our method may not optimally represent dynamically changing tracks.

However, we are confident that our work still provides valuable insights into the spatial organization of the microtubule network, and the results obtained are reliable for the majority of the observed transport paths.

As suggested by the referee, we have added a brief comment to the revised manuscript to acknowledge this limitation as follows:

(Before) In lines 9-10 on page 7, *“Thus, the dynamic imaging with the extremely broad bandwidth demonstrated here is unprecedented in fluorescence or vibrational microscopies”*.

(After) *“Thus, the dynamic imaging with the extremely broad bandwidth demonstrated here is unprecedented in fluorescence or vibrational microscopies.*

In this study, we determined the locations of intracellular cargos using cargo-localization iSCAT microscopy, combined with image processing methods that extract the dynamic features of moving cargos from the complex cytoplasmic landscape. However, continuous tracking of label-free cargos is limited in duration, as image processing methods tailored for dynamic entities (e.g., TD-iSCAT) may lose track of them when they become momentarily immobile. Additionally, measuring cargo locations can be challenging when cargos overlap with other similar cytoplasmic objects or when their scattering

signal is significantly overshadowed by unwanted signals from unspecified cellular environments. While the underlying active cytoskeletal structure can be identified using cargo-localization iSCAT microscopy, the reconstruction process requires a substantial volume of data, comprising thousands of consecutive image frames taken over an extended period (~ 3 min). Consequently, tracking the temporal dynamics of cytoskeleton remodeling on shorter timescales remains unattainable. Furthermore, while state-of-the-art iSCAT technologies enable tracking the Brownian motion of nanoparticles in a three-dimensional space, accurately measuring the variation in their vertical positions within the highly crowded and inhomogeneous cytoplasm remains challenging, due to the presence of numerous optically heterogeneous cytoplasmic objects. Consequently, the cargo-localization density map constructed by cargo-localization iSCAT microscopy should be considered a 2D projection of all cargos moving in 3D within the depth of focus of our microscope.”

(Q13) Fig 2: what do the green arrows mean in ‘a’? nothing is mentioned about it.

(A13) The green arrowheads drawn in Figs. 2a and b indicate some fluctuating debris outside the cell, which was mentioned in lines 13-14 in page 13. Because these debris were thermally fluctuating, they were detected in TD-iSCAT. It also appeared in the density map.

(Q14) Fig 3: (e): is f) showing the positions of the trajectory in e). If so, the x and y axis graphs should both go from 0 to negative values. Because the trajectory goes from the top right to bottom left corner. If f) depicts another data set, it would be more insightful if the corresponding track is shown in e).

(A14) We are grateful to the referee for their meticulous examination of our manuscript, which allowed them to identify this elusive error. In accordance with the referee's indication, we have revised Fig. 3 (e), where the y-axis is reversed. This has also been reflected in Fig. 3 (f), where the displacements of the cargo along the x- and y-axis have been corrected to align with the revised axis direction (Rebuttal_Fig. 11). Additionally, we have addressed another (unnoticed) error in the trajectory drawn in Fig. 4 (b) by reversing the direction of x-axis in the revised manuscript (Rebuttal_Fig. 12). Thank you for bringing these issues to our attention, and we believe that the revised figures will improve the clarity and accuracy of our results.

Rebuttal_Fig. 11. (left) Fig. 3e, f in the original manuscript, (right) its corrected version in the revised manuscript.

Rebuttal_Fig. 12. (top) Fig. 4b in the original manuscript, (bottom) its corrected version in the revised manuscript.

(Q15) Page 19, line 35 till page 20, line 2 : This has nothing to do in a methods section.

(A15) Thank you for bringing this to our attention. The sentences in question were used to explain two different types of iSCAT image processing methods applied in Figs. 1c, d of the original manuscript, and therefore would be more appropriately placed in the main text. We have revised the manuscript and moved this information to a more suitable location in the text.

(Before) In lines 8-10 on page 3, “*To discern mobile cargos from such static objects and sort out directional cargos from random walkers, we employed the time-differential iSCAT (TD-iSCAT) method (Fig 1d, Extended Data Fig. 2b, d)*”.

(After) “*To discern mobile cargos from such static objects and sort out directional cargos from random walkers, we employed the time-differential iSCAT (TD-iSCAT) method (Fig 1d, Supplementary Fig. 2b, d). In the case of a directionally moving cargo, the dipolar appearance of the cargo reveals its moving direction, indicated by the arrow, which points from dark to bright spots Supplementary Fig. 2d). In contrast, the corresponding SBR-iSCAT images fail to reveal the cargo, as the irregular cytoplasmic background interferes with its detection (Supplementary Fig. 2c). The TD-iSCAT image sequence displays the cargo vanishing instantaneously at $t = 1.96$ s (Supplementary Fig. 2d) when it altered direction, implying a brief pause on a cytoskeletal tracks. The observed separation between the bright and dark spots represents the displacement of the cargo over time interval Δt . The cargo's movement resembles that of a manufactured vehicle, as it slows down, stops, turns around, and speeds up in the opposite direction (Supplementary Fig. 2d). For backward motion, cellular cargos have two options: engaging motor proteins with opposite directionality on the same track or, less likely, using another track with opposite polarity.*”

Referee #3

(Overall remark) The authors present a new method for tracing features of the cytoskeleton in live cells. To do so, they combine interferometric scattering microscopy (iSCAT) with differential frame analysis. As cargo attached to cytoskeletal filaments move, they can be identified on top of an otherwise busy background. In a sense, this clever label-free strategy shares some aspects of fluorescence super-resolution microscopy such as PAINT, except that instead of using a large number of bleaching dye molecules, one exploits indigenous biological cargo. This is only made possible because of the high sensitivity of iSCAT for detecting small nanoparticles.

Although the authors do not address or solve any particular biological question, they show that their method could be used to investigate cargo diffusion and transport, which is a very important and complex matter in cell biology. I, therefore, deem this work worthy of attention of a wider readership. However, there are a number of critical questions that would have to be addressed before the work reaches the required quality for publication in Nature Communications:

(Reply) We are grateful to the referee for acknowledging the significance of our research. We have included a point-by-point response to their comments below.

(Q1) The authors should refrain from using formulations such as “surprising” or “amazement” when referring to cargo trafficking or other phenomena unless they can provide hard evidence that the observation at hand is surprising, i.e. contrary to what is expected in the literature. They might mean to say “intriguing” instead?

(A1) We appreciate the referee's suggestion to improve the wording used in the original manuscript. Based on their feedback, we have made the following revisions to the manuscript as outlined below.

(Before) In the line 16 on page 7, “*Surprisingly, a cell has inherent strategies for efficient cargo delivery to*”.

(After) “*According to our observations, a cell has inherent strategies for efficient cargo delivery to*”.

(Before) In the line 27 on page 1, “*To our amazement, cells intrinsically have an efficient transport strategy to*”.

(After) “*Intriguingly, cells intrinsically have an efficient transport strategy to*”.

(Q2) The method proposed would only work if the cytoskeletal feature under examination does not change within the measurement time, which seems to be several minutes at times. The authors should make a statement about the limitations of their technique in this regard.

(A2) We acknowledge the referee's point on the limitation of our current CL-iSCAT imaging method. In response to the referee's comment above, we briefly summarize the limitations in the cargo-localization iSCAT method in ‘Conclusion’ of the revised manuscript as below.

(Before) *In lines 9-10 on page 7, “Thus, the dynamic imaging with the extremely broad bandwidth demonstrated here is unprecedented in fluorescence or vibrational microscopies”.*

(After) *“Thus, the dynamic imaging with the extremely broad bandwidth demonstrated here is unprecedented in fluorescence or vibrational microscopies.*

In this study, we determined the locations of intracellular cargos using cargo-localization iSCAT microscopy, combined with image processing methods that extract the dynamic features of moving cargos from the complex cytoplasmic landscape. However, continuous tracking of label-free cargos is limited in duration, as image processing methods tailored for dynamic entities (e.g., TD-iSCAT) may lose track of them when they become momentarily immobile. Additionally, measuring cargo locations can be challenging when cargos overlap with other similar cytoplasmic objects or when their scattering signal is significantly overshadowed by unwanted signals from unspecified cellular environments. While the underlying active cytoskeletal structure can be identified using cargo-localization iSCAT microscopy, the reconstruction process requires a substantial volume of data, comprising thousands of consecutive image frames taken over an extended period (~ 3 min). Consequently, tracking the temporal dynamics of cytoskeleton remodeling on shorter timescales remains unattainable. Furthermore, while state-of-the-art iSCAT technologies enable tracking the Brownian motion of nanoparticles in a three-dimensional space, accurately measuring the variation in their vertical positions within the highly crowded and inhomogeneous cytoplasm remains challenging, due to the presence of numerous optically heterogeneous cytoplasmic objects. Consequently, the cargo-localization density map constructed by cargo-localization iSCAT microscopy should be considered a 2D projection of all cargos moving in 3D within the depth of focus of our microscope”.

(Q3) The authors do not reveal any critical cell biology information. Instead, their evidence on transport phenomena remains at the speculative level. The authors are encouraged to demonstrate if the observations are representative of what happens on the cellular level.

(A3) We acknowledge the referee's concerns and have taken the necessary steps to provide further evidence to support our findings. In response to the referee's comments, we have performed additional experiments to demonstrate the relationship between intracellular traffic flows observed by iSCAT microscopy and cargo transportation on the microtubule network in eukaryotic cells. These experiments include:

- (i) Simultaneous imaging of fluorescently-labeled microtubules and intracellular traffic flows (Rebuttal_Fig. 8 or Supplementary Fig. 10 in the revised manuscript),
- (ii) Assessment of the impact of the microtubule-targeting drug, nocodazole, on intracellular traffic flows (Rebuttal_Fig. 9 or Supplementary Fig. 11 in the revised manuscript), and
- (iii) Simultaneous tracking of specifically labeled late endosomes with unlabeled cargos (Rebuttal_Fig. 10 or Supplementary Fig. 12 in the revised manuscript).

These results provide strong evidence of the critical role that microtubules play in cellular transport and support our main findings that intracellular traffic phenomena are a common feature in cellular transport, primarily associated with the microtubule network.

Rebuttal_Fig. 8 (or see **Supplementary Fig. 10** in the revised manuscript). **Dual-channel F-iSCAT imaging captures the intracellular transport network of a COS-7 cell.** **a**, Fluorescence image displaying the spatial distribution of microtubules labeled with GFP. **b**, SBR-iSCAT and **c**, TD-iSCAT snapshots corresponding to the fluorescence image. Grayscale legends in (b) and (c) are used as in Fig. 1c and 1d, respectively. **d**, Magnified fluorescence image of the red-boxed area in (a). This area highlights seven microtubule intersections, marked by blue circles. **e**, The cargo-localization density map was created from 15,000 consecutive iSCAT images taken at 50 Hz, showing microtubule intersections as concentrated areas of cargo (white circles). Here, color legend is used as in Fig. 2c except for using 15,000 frames ($\Delta T = 300$ s). **f**, 402 trajectories of individual cargos superimposed onto the magnified TD-iSCAT image of the red-boxed area in (c) displays the active transport routes through the microtubule network. The white circles in (e) and (f) that correspond to the microtubule intersections marked by blue circles in (d) also exhibit high density regions in the density map in (e) and serve as primary passageways for cargo transportation where multiple cargo trajectories intersect with each other as shown in (f). The cargos selected here are among those tracked continuously over 1 s.

Rebuttal_Fig. 9 (or see **Supplementary Fig. 11** in the revised manuscript). **Impact of nocodazole treatment on intracellular traffic in a COS-7 cell.** **a**, SBR-iSCAT snapshots of a COS-7 cell before and **b**, after treatment with 2 μM nocodazole. Nocodazole was treated at $t = 20$ min. Grayscale legend is used as in Fig. 1d. **c**, Locations of dynamic cargos (orange circles) identified from TD-iSCAT snapshots (yellow-boxed area in (a, b)) taken at two different time point, $t = 0$ and 28 min. **d**, Sequential image of cargo localization density maps of the yellow-boxed area in (a, b). Each image was generated from 9,000 consecutive frames taken at 50 Hz. Supplementary Video 5 and 6 show cargo dynamics before and after nocodazole treatment, respectively. In (d), color legend is used as in Fig. 2d. Scale bars: 2 μm .

Rebuttal_Fig. 10 (or see **Supplementary Fig. 12** in the revised manuscript). **Imaging of GFP-labeled late endosomes (LEs) in a COS-7 cell using F-iSCAT microscopy.** **a**, Fluorescence image showing 14 LEs marked by numbered yellow circles. **b**, SBR-iSCAT and **c**, TD-iSCAT images corresponding to (a). Yellow circles with the same numbers in (a) indicate co-localized cargos, *i.e.*, LEs detected in TD-iSCAT image, while red circles indicate other cargos only detected in TD-iSCAT image. Grayscale legends in (b) and (c) are used as in Fig. 1c and 1d, respectively. **d**, Trajectories of 14 LEs numbered as in (a) and tracked at 10 Hz using fluorescence microscopy, depicted in different colors. (Inset: typical snapshots of LE (#4) captured by fluorescence (F), SBR- and TD-iSCAT images.) **e**, Cytoskeleton network reconstructed from TD-iSCAT images using cargo localization, based on 15,000 consecutive frames taken at 50 Hz. Four representative trajectories of LEs (#2, #3, #4, and #14) from (d) are overlaid. **f**, Traffic density map showing the total number of cargos detected at each pixel for a total of 15,000 consecutive frames ($\Delta T = 300$ s), indicated by the color legend. See also Supplementary Video 3.

(Q4) Considering that the traced cytoskeletal filaments cannot perfectly lie in a horizontal plane, the authors should make a statement about the consequences of axial motion, which changes the iSCAT contrast.

(A4) We acknowledge the referee's concerns about the contrast change due to the axial motion of a cargo. The region of lamellipodium and lamella analyzed in this study appears flat and quasi-2D, however, as pointed out by the referee, the cytoskeletal network in living cells has a complex 3D architecture at the nanoscale. The iSCAT technique offers the possibility of 3D particle tracking with nanometer precision and at high speeds up to microseconds per frame, but these strengths are typically limited to relatively simple external membrane surfaces (11-13). Our cargo-localization density map presented in the original manuscript represents a 2D projection of all cargos moving on the 3D cytoskeletal network. In response to the referee's comment, we have added a paragraph in the revised manuscript to summarize the limitations of iSCAT when applied to a heterogeneous and three-dimensional living cell (please see our answer to Q2 above). We have also included Supplementary Fig. 3 (or Rebuttal_Fig. 1) to clearly show that the iSCAT contrast in the tracking of a dynamic cargo varies continuously, reflecting its axial motion along the 3D cytoskeletal network (Rebuttal_Fig. 1c). We have

summarized the limitations of the current cargo-localization iSCAT method such as the difficulty in vertical tracking in heterogeneous cellular environments in our answer (A2) above.

Rebuttal_Fig. 1 (see **Supplementary Fig. 3** in the revised manuscript). **Continuous cargo tracking with SBR- and TD-iSCAT methods.** **a**, Trajectories of a GFP-labeled late endosomal cargo, revealed by fluorescence detection (left, white) and SBR-iSCAT (middle) or TD-iSCAT (right) imaging-based cargo localization. The trajectories by iSCAT are color-coded based on the elapsed time and superimposed on a snapshot taken at $t = 0.0$ s. Color legend is used as in Fig. 1g. **b**, Comparison of the two trajectories of the cargo obtained by SBR- and TD-iSCAT methods (blue: SBR-iSCAT; red: TD-iSCAT) highlighting the complementarity of the two methods for persistent cargo tracking. Only the y-positions of the cargo from the trajectories are plotted for ease of comparison. Three representative moments when the cargo was detected by either TD-iSCAT only, SBR-iSCAT only, or both, marked as (i), (iii), and (ii) respectively, are captured by snapshots (left: SBR-iSCAT; right: TD-iSCAT; yellow circle: detected cargo). The trace of the cargo with SBR-iSCAT was successfully acquired except for a few moments where it was masked by an unknown object (case (i)). The cargo was frequently lost in TD-iSCAT due to its intermittent pauses (case (iii)). **c**, The SBR-iSCAT contrast of the cargo, reflecting its axial motion, is shown with images of the cargo in the brightest and darkest contrast next to a peak and a trough in the time trace, respectively. See also Supplementary Video 3.

[References]

11. Taylor, R. W. et al. Interferometric scattering microscopy reveals microsecond nanoscopic protein motion on a live cell membrane. *Nat. Photon.* **13**, 480-487 (2019).
12. de Wit, G., Albrecht, D., Ewers, H. & Kukura, P. Revealing compartmentalized diffusion in living cells with interferometric scattering microscopy. *Biophys. J.* **114**, 2945–2950 (2018).

13. Huang, Y.-F., Zhuo, G.-Y., Chou, C.-Y., Lin, C.-H., & Hsieh, C.-L. Label-free, ultrahigh-speed, 3D observation of bidirectional and correlated intracellular cargo transport by coherent brightfield microscopy. *Nanoscale* **9**, 6567–6574 (2017).

(Q5) Although I find the author's analogies with urban traffic interesting, I believe such continuous emphasis would only be warranted if it could go beyond an anecdotal level and were based on a quantitative statistical analysis. I recommend they remove those references.

(A5) We appreciate the referee's feedback on our use of analogies with urban traffic to explain intracellular transport phenomena. We understand the importance of providing a quantitative basis for these analogies and agree that they should be based on a statistical analysis.

Indeed, the analogy between urban traffic and cellular transport is both reasonable and natural, as the fundamental elements in each system share striking similarities (14,15). Cellular transport involves the movement of cargo via motor proteins (such as kinesins, dyneins, and myosins) that travel directionally along a proteinaceous cytoskeletal network (composed of microtubules and actin filaments). This is remarkably similar to motor vehicles navigating directionally on paved roads in urban transport systems. Given these parallels, it is logical to expect a wide range of features, characteristics, and events to exhibit similarities or even fundamental equivalencies across both systems. When considering active movers traveling along a virtually 1D track, collisions and temporary stalls become inevitable due to the lack of alternative paths. However, a notable difference is that motor proteins occasionally dissociate from the track into the cytosol, an event not possible in man-made traffic. In both types of transport, mobility is significantly reduced when movers are concentrated in a confined area, leading to traffic jams.

We find it particularly intriguing to examine the unusual cases that occur in both systems. For instance, strategies to overcome congestion can be observed in cellular transport as well as urban traffic, where cargos change tracks or turn around to avoid collisions. When two cargos meet, the one arriving later typically waits for the one arriving earlier to pass the junction, which is an expected behavior. To improve efficiency, we have observed that cells often transport materials in packets, similar to how goods are delivered in our socio-economic systems. Dimeric cargos are frequently found, and even larger multimers have been routinely observed, akin to trucks carrying long trailers. Hitchhiking or free riding is also found in cellular transport, as observed in our society. Additionally, traffic flow surges in response to special events, such as the extension of filopodia at the cell periphery, which is reminiscent of human logistics behavior. Overall, these observations support the idea that the urban traffic analogy is both reasonable and natural in the context of cellular transport.

In response to the referee's concerns, we have added quantitative statistical data to support our analogies in the revised manuscript. We have included information on the frequency of dimeric cargos observed in TD-iSCAT imaging (Rebuttal_Fig. 7, or Supplementary Fig. 14). Therefore, we respectfully request that we be allowed to maintain our current analogical picture in the revised manuscript.

[References]

14. Liberali, P. Snijder, B. & Pelkmans, L. A hierarchical map of regulatory genetic interactions in membrane trafficking. *Cell*. 157, 1473-1487 (2014). Also, see the news article published as "Cellular traffic control system mapped for the first time" in Phys.org/news/2014-06-cellular-traffic.html, provided by Univ. Zurich.

15. Leduc, C. et al. Molecular crowding creates traffic jams of kinesin motors on microtubules. *Proc. Natl. Acad. Sci. U. S. A.* 109, 6100-6105 (2012). Also, see the commentary written by Ross, J. L. The impacts of molecular motor traffic jams, *Proc. Natl. Acad. Sci. U. S. A.* **109**, 5911-5912 (2012).

(Q6) Page 4, lines 28-32: Can the authors say more about the fact that their measurements are more precise than previous ones?

(A6) Thank you for raising this point. We agree that the precision of our measurements is indeed a crucial aspect of our study. Our measurements were performed using the iSCAT technique, which has the advantage of providing a higher localization precision compared to fluorescence microscopy, which was used in many previous studies. This is due to the fact that iSCAT has a higher photon budget, which results in higher scattered light intensity compared to fluorescent light intensity. As a result, our measurements can provide a more accurate and detailed representation of intracellular transport phenomena. However, it is important to note that our measurements may be biased towards more dynamic cases since we only detect dynamic cargos with TD-iSCAT and may fail to capture very slow cargos. Therefore, our distributions of motor speed may be narrower compared to previous studies.

(Q7) The method would only be truly valuable if it could be implemented without synthetic cargo such as PS beads. The authors should make a clear statement/demonstration of biological information that can be obtained from indigenous cargo.

(A7) We appreciate the referee's feedback on the importance of demonstrating the biological information that can be obtained from indigenous cargo. As stated in our original manuscript, TD-iSCAT microscopy was specifically designed to study the movement of native intracellular cargos along the cytoskeleton. While we used f-PS beads as an indicator to validate our label-free imaging of dynamic cargos, we acknowledge that the true value of our method lies in its ability to study native cargo without the use of synthetic particles. To address the referee's comment, we have added new data to the revised manuscript that demonstrates the ability of TD-iSCAT to study traffic flows in COS-7 cells without f-PS beads (Rebuttal_Fig. 8 or Supplementary Fig. 10 in the revised manuscript) or with other GFP-labeled biological targets, such as microtubules and late endosomes (Rebuttal_Figs. 8 and 10, or Supplementary Figs. 10 and 12, respectively). These new data highlight the versatility and robustness of TD-iSCAT in studying biological transport processes in living cells.

REVIEWERS' COMMENTS

Reviewer #1 (Remarks to the Author):

I believe the revised manuscript sufficiently addresses the concerns about the original submission. In particular, the authors provide substantial new data on

- (1) reconstruction of microtubule networks based on particle trajectories, as well as effects of nocodazole on trajectories
- (2) long-time hitch-hiking runs and u-turns that distinguish linked hitchhiker particles from nearby cargos running in sync
- (3) Using 2 methods simultaneously to track particles through different phases of motion, and to track fluorescent particles together with unlabeled ones.

While the new data is still largely descriptive rather than quantitatively analyzed, I believe the method is sufficiently novel and powerful to be worth publishing and making available to a broad readership. The availability of the large datasets of long trajectories would be a boon to other groups interested in quantifying intracellular particle movements.

Reviewer #2 (Remarks to the Author):

The authors have taken some commendable steps to address the reviewers comments. One important addition are the fluorescence microscopy images of microtubules to demonstrate that the majority of iSCAT tracks of vesicles correlate with the position of microtubules. The authors now also provide more examples of vesicle behaviour and dynamics to underline their statements. However, a thorough statistical analysis, e.g. what percentage of tracks show a U-turn or a jamming event, remains missing which limits the strength of the claims.

The authors claim that the iSCAT approach is novel and unique to gather large amounts of vesicle tracks using a label free approach. However, a paper that is even cited by the authors (#26 in the manuscript) presents a very similar approach, COBRI, by which cellular vesicles can be tracked label free at 30000

frames per second with the advantage of 3D tracking. At least, the authors should discuss how their approach differs from the COBRI method.

In summary, the paper is still very descriptive and reads more like a methods paper. And for a methods paper, it lacks further characterization, e.g. how the selection of contrast threshold levels affects the reliability of particle tracking, are smaller particles lost because of that? How well does the TD-iSCAT work on larger, slow-moving particles?

additional comments:

-the authors make a comment that the shape of tracks suggest that these are along microtubules with a persistence length of 4-8 μm .

As mentioned in the previous review report, the persistence length of microtubules in cells is not 4 μm but shorter. Observing active fluctuations is not the only way to deduce the effective persistence length, another is to measure the abundance of curvatures (one recent example of this established method is DOI: 10.1016/j.bpj.2022.04.020). As the authors propose to use the cargo tracks to reconstruct the microtubule network, it would be a good validation to compute the persistence length of the tracks. It would be wrong to argue that the observed structures represent filaments with a persistence length of 4-8 μm , because then, they should not see any meaningful bends on a cellular length scale.

in addition, I don't understand why there is such an emphasis on the analogy to car traffic. There are no bundles of microtubules leading from the center of the cell to the periphery, it's rather an array of many individual microtubules that lead towards different positions at the cell edge. This is very different to a highway network.

in contrast to cars, vesicles can move along the top, the bottom or the side of a microtubule. Multiple types of motors can connect to the same vesicle, etc. I understand that road systems are often used as an analogy to cargo traffic, but I would use it less heavily as this simple picture has its limitations.

Reviewer #3 (Remarks to the Author):

Park et al have substantially improved their manuscript by presenting new data and addressing many of the issues raised by the referees. In particular, combination with fluorescence measurements helps a great deal in substantiating the claims based on the iSCAT results. Although the biological data and conclusions remain elementary, I believe the authors show in a convincing manner the value of their imaging and analysis strategies for obtaining new insights that serve as complementary tools for cellular imaging and quantitative studies. Although I did not examine the comments of the other referees and

the authors' responses to them in great detail, I have the impression that these have been addressed just as my own concerns.

In conclusion, I find the current manuscript worthy of publication in Nature Communications because it introduces a new simple approach for investigating cargo motion in live cells. However, the authors should cite a paper by Kueppers, et al, which has recently appeared in Nature Communications, reporting on the use of confocal iSCAT for label-free imaging of live cells. This new publication can be cited in the introductory part of the manuscript and would be relevant to the new passage that starts with "In this study, we determined the locations of intracellular cargos...". In particular, the statement that "Consequently, tracking the temporal dynamics of cytoskeleton remodeling on shorter timescales remains unattainable." should be revised to account for the achievements of Kueppers et al. I emphasize though that although confocal iSCAT can also be used for studying the phenomena of concern by Park et al, the current manuscript remains very valuable because it shows that one can extract sufficiently high-quality data even without the use of 3D confocal imaging.

REVIEWERS' COMMENTS

Reviewer #1 (Remarks to the Author):

(General remarks) I believe the revised manuscript sufficiently addresses the concerns about the original submission. In particular, the authors provide substantial new data on

- (1) reconstruction of microtubule networks based on particle trajectories, as well as effects of nocodazole on trajectories
- (2) long-time hitch-hiking runs and u-turns that distinguish linked hitchhiker particles from nearby cargos running in sync
- (3) Using 2 methods simultaneously to track particles through different phases of motion, and to track fluorescent particles together with unlabeled ones.

While the new data is still largely descriptive rather than quantitatively analyzed, I believe the method is sufficiently novel and powerful to be worth publishing and making available to a broad readership. The availability of the large datasets of long trajectories would be a boon to other groups interested in quantifying intracellular particle movements.

(Answer) We are delighted to hear that Reviewer #1 believes our revised manuscript has sufficiently addressed the concerns raised about the original manuscript. We sincerely appreciate their time, effort, and positive evaluation of our manuscript.

Reviewer #2 (Remarks to the Author):

(General remarks) The authors have taken some commendable steps to address the reviewers comments. One important addition are the fluorescence microscopy images of microtubules to demonstrate that the majority of iSCAT tracks of vesicles correlate with the position of microtubules. The authors now also provide more examples of vesicle behaviour and dynamics to underline their statements. However, a thorough statistical analysis, e.g. what percentage of tracks show a U-turn or a jamming event, remains missing which limits the strength of the claims.

The authors claim that the iSCAT approach is novel and unique to gather large amounts of vesicle tracks using a label free approach. However, a paper that is even cited by the authors (#26 in the manuscript) presents a very similar approach, COBRI, by which cellular vesicles can be tracked label free at 30000 frames per second with the advantage of 3D tracking. At least, the authors should discuss how their approach differs from the COBRI method.

In summary, the paper is still very descriptive and reads more like a methods paper. And for a methods paper, it lacks further characterization, e.g. how the selection of contrast threshold levels affects the reliability of particle tracking, are smaller particles lost because of that? How well does the TD-iSCAT work on larger, slow-moving particles?

(Answer) We appreciate the valuable feedback provided by Reviewer #2. We have carefully considered the reviewer's comments and have revised our manuscript to address the concerns raised. Below, we provide a point-by-point response to this reviewer's comments.

(1) Statistical analysis of specific events such as a 'U-turn' and 'jamming': First and foremost, we would like to emphasize that, in response to the reviewer's request on our original manuscript, we have incorporated a new dataset of the 'U-turn' event (Supplementary Fig. 18 in the revised manuscript), which provides clear experimental evidence of a hitchhiking cargo. We believe that the inclusion of additional datasets such as the 'U-turn' of a dimeric cargo (Supplementary Fig. 19 in the revised

manuscript) and the trajectories of eight dimeric cargos (Supplementary Fig. 15 in the revised manuscript) indicates that cargo transport in a dimeric form is not a rare occurrence in a crowded cellular environment. The demonstration of the U-turn event is integral in providing evidence for a physically bound dimeric cargo. Consequently, the emphasis is on the occurrence of such an event, rather than its frequency. While we didn't quantify the frequency of U-turn events in the updated manuscript, there is no question that we would identify an increased number of these events when we examine a larger volume of image data. Furthermore, we acknowledge the technical challenges posed by manual identification and tracking of dimeric cargos in a crowded cellular environment. The scattering signals from these cargos are frequently disturbed by interference with other cellular obstacles, resulting in frequent loss of their locations during tracking. To address these challenges effectively, it is imperative to develop an advanced particle tracking tool that leverages machine learning for automatic identification and long-term tracking of dimeric cargos. Adopting this innovative approach would undoubtedly prove highly beneficial in overcoming the limitations associated with manual tracking.

In accordance with the reviewer's request on the original manuscript, we have demonstrated that the number of cargos in a traffic jammed area can be quantified using iSCAT microscopy (Supplementary Fig. 14 in the revised manuscript). Moreover, we have shown that the traffic density of cargos is indeed higher at the intersection of microtubule networks as revealed by dual-channel F-iSCAT imaging (Supplementary Fig. 11 in the revised manuscript).

While we have not detailed individual event statistics in our observations of cargo traffic, we are confident that the additional evidence presented in our revised manuscript convincingly demonstrates the representativeness of these phenomena. The cargo traffic behavior observed through iSCAT microscopy, we argue, adequately depicts the broader cellular dynamics, transcending merely anecdotal observations and offering insights applicable to general phenomena.

Novelty of iSCAT imaging compared to the coherent brightfield (COBRI) technique: We appreciate the reviewer's comment regarding the novelty of our iSCAT imaging method compared to the COBRI technique. The COBRI technique is akin to iSCAT; however, it serves as the transmission version, detecting signals transmitted through the sample, while iSCAT, its forerunner, picks up signals reflected from the sample. While both methods have no significant difference in detection sensitivity and imaging speed in principle, they capture cellular structures in distinct ways. In the SBR-iSCAT images, various subcellular organelles are clearly visible with strong fringes generated from the plasma membrane surface. These cellular structures appear almost transparent in COBRI. This near transparency in COBRI poses challenges when investigating the interactions of cargos within their local surrounding environment. The novelty of our approach lies in the application of the cargo-localization method using iSCAT microscopy to study intracellular traffic phenomena. By leveraging the high detection sensitivity of iSCAT and the ability to visualize cargo behavior in the absence of labels, we have uncovered important insights into cellular traffic problems. While COBRI may have potential for similar applications, its specific capabilities and limitations within this context remain to be explored.

We would like to address the difference in imaging speed between our method and COBRI. It is important to note that this disparity does not arise from an essential difference between the two imaging techniques but is entirely determined by the camera performance and scanned area. In the COBRI study, the authors achieved a high-speed imaging rate of 30,000 Hz by employing a high-speed camera and scanning a smaller view-field area of $6 \times 6 \mu\text{m}^2$. On the other hand, our iSCAT microscopy utilized an sCMOS camera in an x - y scanning mode over a larger view-field area of $20 \times 20 \mu\text{m}^2$, resulting in a maximum speed of ~ 200 Hz.

In this final revision, we have added the advanced technique of confocal iSCAT microscopy, independently developed by Kueppers et al.¹ and by Hsiao et al.² to our manuscript as they hold significant promise for exploring the three-dimensional phenomenon of the cargo traffic along complex cytoskeleton networks as follows:

(Before) In lines 13-20 of page 9 in the manuscript, “*Consequently, the cargo-localization density map constructed by cargo-localization iSCAT microscopy should be considered a 2D projection of all cargos moving in 3D within the depth of focus of our microscope*”.

(After) “*Consequently, the cargo-localization density map constructed by cargo-localization iSCAT microscopy should be considered a 2D projection of all cargos moving in 3D within the depth of focus of our microscope. Recently, confocal-type iSCAT microscopies have been developed, offering the ability to investigate nanoscopic structures and dynamics in living cells^{50, 51}. With its high-resolution detection capability, this imaging technique shows substantial potential in unraveling the intricacies of 3D cargo traffic phenomena along the complex cytoskeleton networks, given its aptitude for detecting nanoscopic displacements in the axial direction*”.

[Newly added References]

1. K ppers, M., Albrecht, D., Kashkanova, A. D., L hr, J. & Sandoghdar, V. Confocal interferometric scattering microscopy reveals 3D nonoscopic structure and dynamics in live cells. Nat. Commun. 14, 1962 (2023). (newly cited as Ref. [50] in the revised manuscript).
2. Hsiao, Y.-T., Wu, T.-Y., Wu, B.-K., Chu, S.-W. & Hsieh, C.-L. Spinning disk interferometric scattering confocal microscopy captures millisecond timescale dynamics of living cells, Opt. Express, 30, 45233-45245 (2022). (newly cited as Ref. [51] in the revised manuscript).

Further characterization of a few technical points that might be required for a methods paper:

The reviewer viewed our paper as a methods paper, which we do not agree to. Our work may appear to be methodological and indeed describes some technical details, but it is because in this work we first introduced a novel localization-based approach of iSCAT. We would emphasize that the theme of our work is to reveal and capture the intricate cargo trafficking in a complex cellular environment. While we acknowledge the importance of further characterization, it is essential to note that our primary objective was to understand the nature of intracellular traffic in a crowded cellular environment using our new imaging method. By employing the label-free cargo localization methodology, we demonstrated that cells employ effective strategies to solve various cellular transportation problems, including traffic congestion.

The level of contrast threshold influences the characterization of iSCAT-based images, which also applies to other imaging techniques such as fluorescence imaging. Therefore, our focus primarily lies in capturing dynamically transported cargos, which are actively moved by motor proteins, rather than displaying various subcellular structures through SBR-iSCAT imaging. In general, an SBR-iSCAT image exhibits a broad distribution in iSCAT contrast (see Rebuttal Figs. 1a and 1c) while the corresponding TD-iSCAT image displays a narrow distribution of contrast near zero (Rebuttal Figs. 1b and 1d). Consequently, when using TD-iSCAT images for cargo localization, relatively static cytoplasmic structures disappear due to signal cancellation regardless of the level of threshold. To identify cargos and track their dynamics in TD-iSCAT images, as explained in the ‘Cargo localization

and tracking via the MOSAIC ImageJ plugin' subsection of the 'Methods' section, we utilized two parameter values: particle radius (r) and intensity percentile (I_p). The r value was set to 7 pixel (~ 280 nm), corresponding to the diffraction limit in our imaging system. I_p was set to 0.3 to ensure the selection of the bright spots, each one of which pair with a dark spot nearby because the appearance of a proximal pair of bright and dark spots is a characteristic feature of dynamic cargo revealed by TD iSCAT images (depicted by white circles in Rebuttal Fig. 1b and insets in Rebuttal Fig. 1f). With a larger value of $I_p = 0.5$, other isolated spots with bright contrast, not paired by a dark spot, began to be detected (indicated by pink circles in Rebuttal Fig. 1b and insets in Rebuttal Fig. 1e). Through further analysis, we found that the chosen value of $I_p = 0.3$ used for cargo identification corresponds to the level of contrast (brightness), which is approximately 4.5 times the standard deviation (σ) above the mean (μ) of the Gaussian distribution of iSCAT contrast obtained from the whole pixels (see Rebuttal Figs. 1d and 1h). Remarkably, all the 11 cargos detected with $I_p = 0.3$ (white circles in Rebuttal Fig. 1b) precisely coincide with the bright spots with the contrast $> \mu + 4.5\sigma$ (white circles in Rebuttal Fig. 1h). Those detected by the two different criteria are not perfectly matched: Three bright spots detected by the same contrast criterion, however, belong to those detected at $I_p = 0.5$ (red circles in Rebuttal Fig. 1h). The condition of the contrast $> \mu + 3.5\sigma$ scores more false positive events ($I_p = 0.5$) without changing the count of correct cargo localization ($I_p = 0.3$).

Some mobile cargos or larger organelles, which move more slowly than motor-protein driven cargos, may not be detected in TD-iSCAT imaging under the conditions used in our study. We are, however, sure that adjusting the value of time interval, Δt , would allow us to detect slowly moving particles. In the current work, we shall not pursue this any further because this topic is beyond the scope of our study.

In this revised manuscript, we have included a detailed description of the criteria used for cargo localization in the 'Cargo localization and tracking via the MOSAIC ImageJ plugin' subsection in 'Methods' section. Also, to better illustrate our methodology, we have also included a new supplementary figure as Supplementary Fig. 20. Our manuscript is modified as follow.

(Before) In lines 15-19 of page 12 in the manuscript, "*The intensity percentile is the parameter to determine which bright pixels are accepted as cargos. While the dynamic cellular features are only highlighted in TD-iSCAT cell images, SBR-iSCAT cell images reveal the whole cellular landscape, including static or randomly moving objects with enhanced iSCAT contrast. Thus, the intensity percentile was set to different values: $I_p = 5$ and 0.3 for SBR- and TD-iSCAT images, respectively*".

(After) " *I_p is the parameter to determine which bright pixels are accepted as cargos. While the dynamic cellular features are only highlighted in TD-iSCAT cell images, SBR-iSCAT cell images reveal the whole cellular landscape, including static or randomly moving objects with enhanced iSCAT contrast. To construct the cargo-localization density map, we selectively detect dynamic cargos, each of which appears as a pair of bright and dark spots in TD-iSCAT images, by setting the value of I_p to 0.3 (Supplementary Fig. 20). SBR-iSCAT images were synchronously examined to detect locally stalled cargos in a traffic jammed area (Fig. 3b) or cargos that are intermittently stalled during a long journey (Supplementary Fig. 3) by applying a different value of I_p to 5*".

Rebuttal Figure 1 (also see **Supplementary Fig. 20** in the final manuscript). Cargo identification in TD-iSCAT images based on the parameter of intensity percentile (I_p) in the MOSAIC ImageJ plugin. **a, b**, SBR-iSCAT image (a) and its corresponding TD-iSCAT image (b). In (b), cargoes were identified with $I_p = 0.5$. Cargoes with $I_p = 0.3$ were marked by white circles. Cargoes, marked by pink circles, were those detected additionally with $I_p = 0.5$. **c, d**, Histograms showing the distribution of iSCAT contrast taken from the whole pixels in the SBR- (a) and TD-iSCAT image (b). In the inset of (d), two dotted lines indicate the locations corresponding to the threshold values, $I_p = 0.5$ and 0.3 . The thresholds set by $I_p = 0.5$ and 0.3 are nearly equivalent to those set by $\bar{i} + 3.5\sigma$ and $\bar{i} + 4.5\sigma$, respectively, with the mean, \bar{i} , and the standard deviation, σ , obtained from Gaussian fitting (red) to the histogram. **e, f**, Cargo-localization density maps reconstructed from 9,000 consecutive TD-iSCAT images using two different parameter values of $I_p = 0.3$ (e) and 0.5 (f). The insets in (e) and (f) represent the TD-iSCAT images of cargoes detected with $I_p = 0.3$ and in the percentile interval from 0.3 to 0.5 , respectively. A proximal pair of bright and dark spots, typical feature of a dynamic cargo detected with $I_p = 0.3$, is clearly visible in the TD-iSCAT image. However, with $I_p = 0.5$, spots with the bright contrast only were additionally detected, indicating that setting a lower threshold likely result in false positive counts. **g, h**, Bright spots (yellow) identified in the TD-iSCAT image in (b) by the level of intensity ($\bar{i} + 4.5\sigma$ and $\bar{i} + 3.5\sigma$), which correspond to $I_p = 0.3$ and 0.5 , respectively. White and pink circles in (g, h) indicate the locations of bright spots identified as cargoes in the TD-iSCAT image with $I_p = 0.3$ and 0.5 , respectively. Other spots, not detected as cargoes in the TD-iSCAT image in (b), were found in the threshold of $\bar{i} + 3.5\sigma$. The SBR-iSCAT image in (a) corresponds to the blue-boxed area in Fig. 1c. Source data are provided in a Source Data file.

Additional comments:

(Q1) The authors make a comment that the shape of tracks suggest that these are along microtubules with a persistence length of 4-8 mm.

As mentioned in the previous review report, the persistence length of microtubules in cells is not 4 mm but shorter. Observing active fluctuations is not the only way to deduce the effective persistence length, another is to measure the abundance of curvatures (one recent example of this established method is DOI: 10.1016/j.bpj.2022.04.020). As the authors propose to use the cargo tracks to reconstruct the microtubule network, it would be a good validation to compute the persistence length of the tracks. It would be wrong to argue that the observed structures represent filaments with a persistence length of 4-8 mm, because then, they should not see any meaningful bends on a cellular length scale.

(A1) We would like to express our gratitude to the reviewer for once again addressing the issue of the persistence length (ξ) of microtubules. In the original manuscript, we mentioned that ξ is in the range of 4 ~ 8 μm , according to earlier works based on *in vitro* assays. Upon careful consideration of the reviewer's comment, we realized that the value of ξ *in vivo* could be significantly shorter, in the range of a few hundred micrometers.

In response to the reviewer's suggestion, we have attempted to calculate the value of ξ of microtubules via the curvature analysis method proposed by Wisanpitayakorn et al⁶. To perform this analysis, we utilized the source code provided by the authors. Three representative cargo traces were obtained by continuously tracking the motion of cargos containing an f-PS bead using fluorescence microscopy at a frame rate of 10 Hz (depicted as empty circles in Rebuttal Figs. 2a-c). During the course of our analysis, we found it challenging to apply their algorithm to our raw data sets (Rebuttal_Figs 1a-c). The algorithm either failed to fit our data points or yielded unreasonably small values of ξ below a few micrometers. We circumvented this difficulty and acquired the value of ξ in a reasonable range (a few hundreds of micrometers as reported by Wisanpitayakorn et al⁶) by fitting (3rd-order) polynomial functions to the raw data points (depicted as red curves in Rebuttal_Figs. 1a-c) and analyzing the curves with the code.

Unlike fluorescence-based imaging, which directly captures the shape of a microtubule filament and thus its local curvature just from a single snapshot of the filament, our cargo-localization approach involves additional uncertainties regarding the shape of the filament because its shape is reconstructed from a large number of positions of cargos transporting along it and the position of a cargo is off from its cytoskeletal foothold by its size. The measurement of cargo position is subject to thermal noise and more seriously, a cargo may rotate about the filament while it is translocated along the microtubule. Such factors add complexities and uncertainties in determining ξ of microtubule. Our preliminary application of the curvature analysis method to the reconstructed shape of microtubule clearly suggests that the value of ξ can be calculated by the method from the data obtained with the cargo-localization method. We believe that a more comprehensive and detailed analysis would be required and should be conducted in future.

We have revised our manuscript and included one new supplementary figure to update the value of ξ of microtubule and to describe our application of the curvature analysis method to estimation of ξ from the reconstructed shape of the filament as follows:

(Before) In the line 23-24 of page 4 in the revised manuscript, "*Their long, straight shape implies that they are built of stiff microtubules with a persistence length of 4 - 8 μm* ^{30,31}"

(After) "*Their long, straight shape implies that they are built of stiff microtubules. From the curvature-based approach reported recently³⁰, we were able to determine the value of persistence length (ξ) of cellular microtubule from its reconstructed shape, which was several hundred micrometers and consistent with previously measured values under *in vivo* conditions (Supplementary Fig. 7)^{30,31}*".

[Newly added Supplementary Figure]

Rebuttal Figure 1 (also see **Supplementary Fig. 7** in the final manuscript). Measurement of the persistence length (ξ) of microtubule through curvature analysis of the shape of a microtubule reconstructed from cargo positions. Three representative trajectories of microtubule were obtained by continuously tracking cargos embedded with an f-PS bead using fluorescence microscopy at a frame rate of 10 Hz (a-c). Raw data points (empty circles) in (a-c) delineate the underlying microtubule's structures. The values of ξ were calculated to be 234, 246, and 289 μm in (a-c), respectively, by analyzing the fitted curves (red) derived from the raw data points using the open-source code provided by Wisanpitayakorn et al³⁰.

Additionally, we replaced the references cited above^{3,4} with the ones more relevant to our live cell experiments following this reviewer's comment^{5,6}.

[Removed references]

3. Gittes, F., Mickey, B., Nettleton, J. & Howard, J. Flexural rigidity of microtubules and actin filaments measured from thermal fluctuations in shape. *J. Cell Biol.* **120**, 923-934 (1993) (cited as (30) in the revised manuscript).

4. Janson, M. E. & Dogterom, M. A bending mode analysis for growing microtubules: evidence for a velocity-dependent rigidity. *Biophys. J.* **87**, 2723-2736 (2004) (cited as (31) in the revised manuscript).

[Newly added references]

5. Wisanpitayakorn, P., Mickolajczyk, K. J., Hancock, W. O. & Tüzel, E. Measurement of the persistence length of cytoskeletal filaments using curvature distributions. *Biophys. J.* **121**, 1813-1822 (2022). (cited as (30) in the final manuscript).

6. Brangwynne, C. P., Koenderink, G. H., Mackintosh, F. C. & Weitz, D. A. Cytoplasmic diffusion: molecular motors mix it up. *J. Cell Biol.* **183**, 583-587 (2008). (cited as (31) in the final manuscript).

(Q2) in addition, I don't understand why there is such an emphasis on the analogy to car traffic. There are no bundles of microtubules leading from the center of the cell to the periphery, it's rather an array of many individual microtubules that lead towards different positions at the cell edge. This is very different to a highway network.

in contrast to cars, vesicles can move along the top, the bottom or the side of a microtubule. Multiple types of motors can connect to the same vesicle, etc. I understand that road systems are often used as an analogy to cargo traffic, but I would use it less heavily as this simple picture has its limitations.

(A2) In response to the reviewer's comment, it's essential to remember that analogies, by nature, serve

to aid in understanding complex concepts by drawing parallels with familiar systems. Our analogy of intracellular transportation to car traffic is not meant to suggest that the two are exactly the same, but rather to provide a simplified, relatable framework for understanding the intricate workings of intracellular transport.

The reviewer mentions the absence of bundled microtubule paths from the cell center to its periphery, unlike a highway network. However, it's important to note that our analogy focuses on the principle of the centralized transportation network, not the exact structural similarities. Just as cars can take various roads to reach different locations in a city, cellular cargos move along different microtubules towards diverse destinations within a cell. There are the central station and transportation hubs in both cellular and urban transportation systems.

Furthermore, the point about vesicles being able to move along all sides of a microtubule adds depth to the analogy rather than undermining it. In real traffic systems, vehicles can change lanes and even reverse direction under certain conditions, much like the vesicles in question.

Multiple types of motors connecting to a single vesicle can be paralleled to a vehicle being driven by different people or automated systems. In essence, while we acknowledge the limitations of the car traffic analogy, its utility in imparting a basic understanding of cellular transportation is undeniable and significant. It is not the exactness, but the relational insight that analogies offer, which makes them an effective pedagogical tool.

Continuing from our previous explanation, our intention was to shed light on the shared complexities both car and intracellular cargo traffic face, and the efficient strategies employed to tackle these hurdles. Drawing upon universally understood issues like traffic congestion, we strove to paint a picture of how cellular elements navigate in densely packed environments. Our ultimate goal was to present a digestible and relatable framework to help readers grasp the intricate dynamics of intracellular cargo transportation. By drawing these parallels, we believe we can stimulate intuitive understanding, thereby facilitating more in-depth exploration of the topic.

Reviewer #3 (Remarks to the Author):

(General remarks) Park et al have substantially improved their manuscript by presenting new data and addressing many of the issues raised by the referees. In particular, combination with fluorescence measurements helps a great deal in substantiating the claims based on the iSCAT results. Although the biological data and conclusions remain elementary, I believe the authors show in a convincing manner the value of their imaging and analysis strategies for obtaining new insights that serve as complementary tools for cellular imaging and quantitative studies. Although I did not examine the comments of the other referees and the authors' responses to them in great detail, I have the impression that these have been addressed just as my own concerns.

In conclusion, I find the current manuscript worthy of publication in Nature Communications because it introduces a new simple approach for investigating cargo motion in live cells. However, the authors should cite a paper by Kueppers, et al, which has recently appeared in Nature Communications, reporting on the use of confocal iSCAT for label-free imaging of live cells. This new publication can be cited in the introductory part of the manuscript and would be relevant to the new passage that starts with "In this study, we determined the locations of intracellular cargos...". In particular, the statement

that "Consequently, tracking the temporal dynamics of cytoskeleton remodeling on shorter timescales remains unattainable." should be revised to account for the achievements of Kueppers et al. I emphasize though that although confocal iSCAT can also be used for studying the phenomena of concern by Park et al, the current manuscript remains very valuable because it shows that one can extract sufficiently high-quality data even without the use of 3D confocal imaging.

(Answer) We extend our sincere gratitude to Reviewer #3 for acknowledging the value of our work and recommending the publication of our manuscript. We would also like to thank the reviewer for bring our attention to a highly relevant and interesting paper. In response to the reviewer's comment, we have incorporated the work by Kueppers et al.⁷ and an independent, much related work by Hsiao et al.⁸ into our revised manuscript as below.

(Before) In lines 17-27 of page 9 in the manuscript, "*While the underlying active cytoskeletal structure can be identified using cargo-localization iSCAT microscopy, the reconstruction process requires a substantial volume of data, comprising thousands of consecutive image frames taken over an extended period (~ 3 min). Consequently, tracking the temporal dynamics of cytoskeleton remodeling on shorter timescales remains unattainable. Furthermore, while state-of-the-art iSCAT technologies enable tracking the Brownian motion of nanoparticles in a three-dimensional space, accurately measuring the variation in their vertical positions within the highly crowded and inhomogeneous cytoplasm remains challenging, due to the presence of numerous optically heterogeneous cytoplasmic objects. Consequently, the cargo-localization density map constructed by cargo-localization iSCAT microscopy should be considered a 2D projection of all cargos moving in 3D within the depth of focus of our microscope*".

(After) "*While the underlying active cytoskeletal structure can be identified using cargo-localization iSCAT microscopy, the reconstruction process requires a substantial volume of data, comprising thousands of consecutive image frames taken over an extended period (~ 3 min). As a result, tracking the temporal dynamics of cytoskeleton remodeling on shorter timescales remains unattainable. Furthermore, while state-of-the-art iSCAT technologies enable tracking the Brownian motion of nanoparticles in a three-dimensional space, accurately measuring the variation in their vertical positions within the highly crowded and inhomogeneous cytoplasm remains challenging, due to the presence of numerous optically heterogeneous cytoplasmic objects. Consequently, the cargo-localization density map constructed by cargo-localization iSCAT microscopy should be considered a 2D projection of all cargos moving in 3D within the depth of focus of our microscope. Recently, confocal-type iSCAT microscopies have been developed, offering the ability to investigate nanoscopic structures and dynamics in living cells^{50, 51}. With its high-resolution detection capability, this imaging technique shows substantial potential in unraveling the intricacies of 3D cargo traffic phenomena along the complex cytoskeleton networks, given its aptitude for detecting nanoscopic displacements in the axial direction*".

[Newly added references]

7. Küppers, M., Albrecht, D., Kashkanova, A. D., Lühr, J. & Sandoghdar, V. Confocal interferometric scattering microscopy reveals 3D nonoscopic structure and dynamics in live cells. Nat. Commun. 14, 1962 (2023). (newly cited as Ref. [50] in the revised manuscript).

8. Hsiao, Y.-T., Wu, T.-Y., Wu, B.-K., Chu, S.-W. & Hsieh, C.-L. Spinning disk interferometric scattering confocal microscopy captures millisecond timescale dynamics of living cells, *Opt. Express*, 30, 45233-45245 (2022). (newly cited as Ref. [51] in the revised manuscript).